# *Salmonella* Typhimurium exploits host polyamines for assembly of the type 3 secretion machinery

**Tsuyoshi Miki**[1]*, **Takeshi Uemura**[2], **Miki Kinoshita**[3], **Yuta Ami**[4], **Masahiro Ito**[1], **Nobuhiko Okada**[1], **Takemitsu Furuchi**[2], **Shin Kurihara**[4], **Takeshi Haneda**[1], **Tohru Minamino**[3]*, **Yun-Gi Kim**[1]

1 Department of Microbiology, School of Pharmacy, Kitasato University, Tokyo, Japan, 2 Laboratory of Bio-analytical Chemistry, Faculty of Pharmaceutical Sciences, Josai University, Saitama, Japan, 3 Graduate School of Frontier Biosciences, Osaka University, Suita, Japan, 4 Faculty of Biology-Oriented Science and Technology, Kindai University, Wakayama, Japan

* mikit@pharm.kitasato-u.ac.jp (TMiki); tohru@fbs.osaka-u.ac.jp (TMinamino)

**Data Availability Statement:** All relevant data are within the paper and its Supporting Information files.

## Abstract

Bacterial pathogens utilize the factors of their hosts to infect them, but which factors they exploit remain poorly defined. Here, we show that a pathogenic *Salmonella enterica* serovar Typhimurium (*S*Tm) exploits host polyamines for the functional expression of virulence factors. An *S*Tm mutant strain lacking principal genes required for polyamine synthesis and transport exhibited impaired infectivity in mice. A polyamine uptake-impaired strain of *S*Tm was unable to inject effectors of the type 3 secretion system into host cells due to a failure of needle assembly. *S*Tm infection stimulated host polyamine production by increasing arginase expression. The decline in polyamine levels caused by difluoromethylornithine, which inhibits host polyamine production, attenuated *S*Tm colonization, whereas polyamine supplementation augmented *S*Tm pathogenesis. Our work reveals that host polyamines are a key factor promoting *S*Tm infection, and therefore a promising therapeutic target for bacterial infection.

## Introduction

Polyamines are aliphatic polycations that are ubiquitous in almost all living organisms, including animals, plants, and microbes [1–3]. Polyamines are chemically characterized by a positive charge at physiological pH that allows electrostatic interaction with the anionic sites of many different macromolecules [4,5]. The interaction of polyamines with their substrates influences various biological reactions, including syntheses of nucleic acids and proteins [3,6]. Thus, it is widely accepted that polyamines are essential for the numerous cellular functions. In mammalian cells, polyamines such as putrescine (PUT), spermidine (SPD), and spermine (SPM) are synthesized and metabolized by an array of enzymatic reactions. In addition, uptake of exogenous polyamines occurs in almost every model organism. Intracellular polyamine contents are strictly regulated in biosynthesis, catabolism, and transport in the cells, and these processes are essential to basic biological reactions such as cell growth and proliferation.

**Funding:** This work was supported in part by Japan society for the promotion of science (JSPS) KAKENHI Grant Number JP21K07030 (to TMiki), JP21K07011 (to NO), JP20K15749 (to MK), JP22K06162 (to MK), JP22K07073 (to TH), JP19H03182 (to TMinamino), JP22H02573 (to TMinamino), JP22K19274 (to TMinamino). YGK is supported by a Fuji Foundation for Protein Research. The funders had no role in study design, data collection and analysis, decision to publish, or preparation of the manuscript.

**Competing interests:** The authors have declared that no competing interests exist.

**Abbreviations:** CI, competitive index; DFMO, difluoromethylornithine; DMEM, Dulbecco's modified Eagle medium; DSS, dextran sulfate sodium; EMA, European Medicines Agency; FDA, Food and Drug Administration; LB, Luria–Bertani; LPM, low in phosphate and magnesium; MOI, multiplicity of infection; NO, nitric oxide; ODC, ornithine decarboxylase; ONPG, o-nitrophenyl-β-D-galactopyranoside; PBS, phosphate-buffered saline; PUT, putrescine; RBC, red blood cell; RNS, reactive nitrogen species; SCV, *Salmonella*-containing vacuole; SMOX, spermine oxidase; SPD, spermidine; SPF, specific pathogen free; SPM, spermine; *S*Tm, *Salmonella enterica* serovar Typhimurium; SRBC, sheep red blood cell; WT, wild-type.

In bacteria such as *Escherichia coli*, PUT and SPD are the predominant polyamines; both are synthesized by a sequence of *spe* gene-encoding enzymes and imported by Pot transporters. PUT is synthesized from SpeC and SpeF. In an alternative route of PUT synthesis, arginine functions as a substrate for SpeA, resulting in production of agmatine, which is metabolized by SpeB into PUT [3]. SPD is then formed from PUT via addition of SpeD-dependent decarboxylated *S*-adenosyl-methionine, mediated by SpeE. In contrast, *E. coli* imports PUT and SPD through 2 distinct transport systems, both characterized as ATP-binding cassette transporters [7–9]. The SPD-preferential uptake system consists of PotA, PotB, PotC, and PotD proteins, whereas PotF, PotG, PotH, and PotI proteins form the PUT-specific uptake system [10–14].

Earlier studies have demonstrated that, in addition to pivotal physiological functions, polyamines are also involved in multiple aspects of bacterial pathogenesis [13,14]. *Salmonella enterica* serovar Typhimurium (*S*Tm) is an entero- and intracellular-pathogenic bacterium causing life-threatening infections ranging from gastroenteritis to systemic infection [15]. The pathogenesis of *S*Tm largely depends on 2 distinct type 3 secretion systems (T3SS-1 and T3SS-2; refer to the Sct common nomenclature [16,17] in S1 Table), in which effectors are secreted and translocated into host cells, allowing *S*Tm to invade into, elicit inflammation in, multiply in, and induce death of host cells [18–21]. Finally, *S*Tm can disseminate to systemic organs such as the liver and spleen via the bloodstream or lymphatic system, leading to colonization of the host and lethal bacteraemia [22]. Polyamines have been reported to contribute to *S*Tm pathogenesis [23–25]; however, the role of polyamines is not fully understood. On the other hand, *S*Tm infection leads to elevated expression of arginase (Arg-1 and Arg-2) [26], which might be a favorable condition for polyamine production. Furthermore, polyamines such as PUT and SPD contribute to *S*Tm virulence by activating the expression of T3SS-1 and T3SS-2 [25] or by inducing a stress response towards reactive oxidative and reactive nitrogen species (RNS) [23]. Similarly, in *Pseudomonas aeruginosa*, SPD has been shown to modulate T3SS expression [27,28]. Very recently, it has become clear that SPD is a key factor in *S*Tm pathogenesis by participating in the early stages of infection, including the adhesion and invasion of *S*Tm into host cells [29]. Here, we clarify that *S*Tm utilizes the polyamines of its hosts in order to infect them. An *S*Tm mutant strain impaired in synthesis and uptake of polyamines exhibited impaired in infectivity in a mouse infection model. Furthermore, polyamine uptake by *S*Tm is necessary for translocation of T3SS effectors into host cells, and thus polyamines are thought to contribute to the assembly of competent T3SS machinery. Therefore, reducing the levels of host polyamines attenuated *S*Tm pathogenesis in a mouse model, whereas SPD supplementation enhanced *S*Tm virulence. Finally, we show that *S*Tm strategically increases host polyamine levels by activating arginase expression, establishing an advantageous condition for the supply of polyamines to *S*Tm for the infection.

## Results

### The uptake plays a critical role in polyamine homeostasis of *S*Tm

The *S*Tm wild-type (WT) strain SL1344 synthesizes PUT and SPD by an array of enzymes encoded by *spe* genes, whereas the polyamines are imported into bacterial cells by PotABCD and PotFGHI, respectively (Figs 1A and S1A). Thus, a polyamine biosynthesis-deficient mutant was constructed by destroying all of the *spe* genes, generating the mutant strain Δ*speABCEDF*. In the transport-deficient mutant, *potAB* and *potFGHI* were deleted, generating the mutant strain Δ*potAB* Δ*potFGHI*. We then constructed 3 biosynthesis- and transport-deficient mutants by deleting *potAB* and/or *potFGHI* in the Δ*speABCEDF* background, generating Δ*speABCEDF* Δ*potAB*, Δ*speABCEDF* Δ*potFGHI*, and Δ*speABCEDF* Δ*potAB* Δ*potFGHI*,

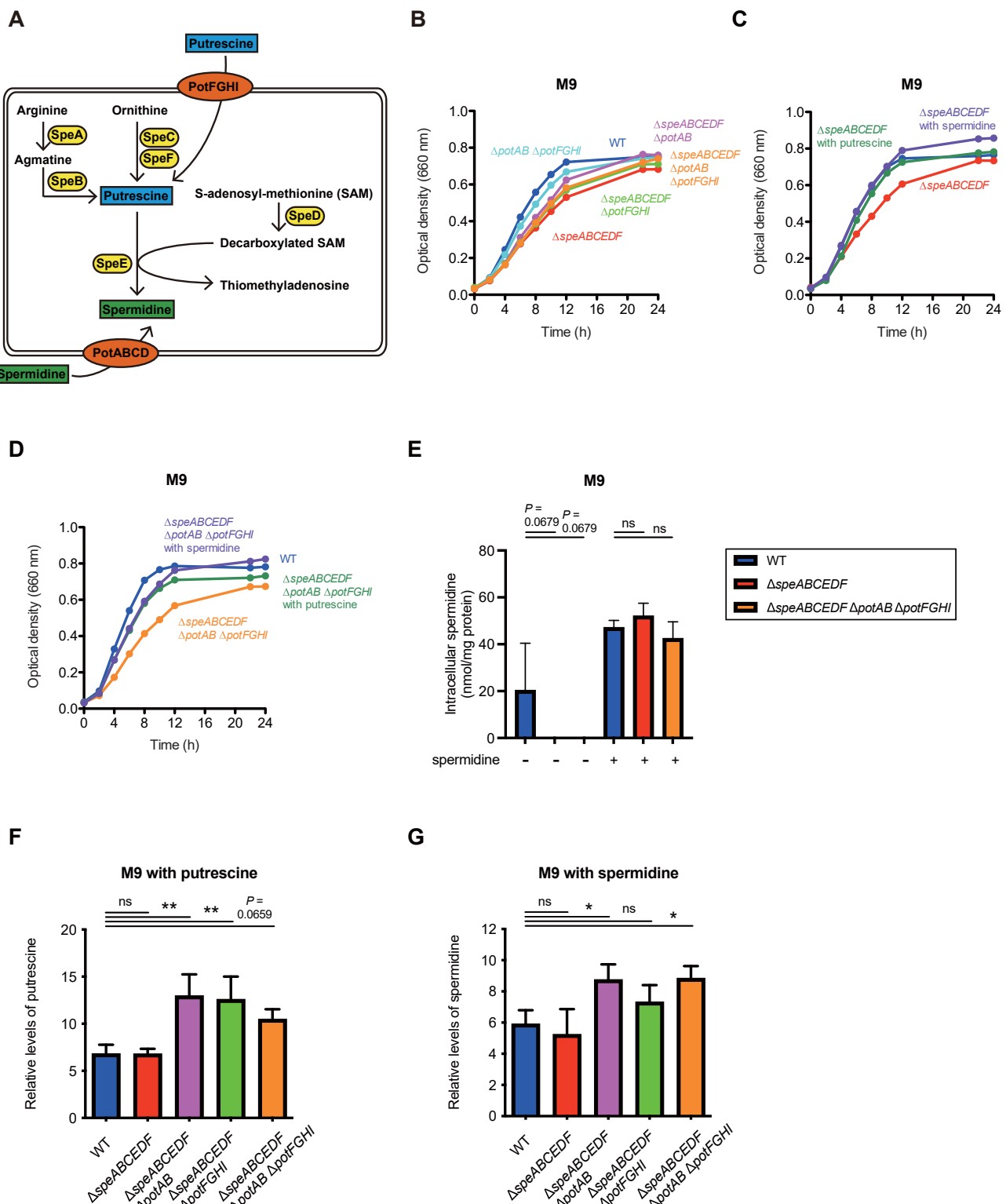

**Fig 1. Polyamine homeostasis is involved in *S*Tm growth in culture media.** (A) Representative polyamine metabolism of *S*Tm. Spe proteins encoded by an array of *spe* genes, and 2 distinct transporters composed of PotABCD and PotFGHI are extensively involved in polyamine homeostasis. (B–D) In vitro growth of *S*Tm WT (blue), Δ*speABCEDF* (red), Δ*potAB* Δ*potFGHI* (sky blue), Δ*speABCDEF* Δ*potAB* (magenta), Δ*speABCDEF* Δ*potFGHI* (light green), and Δ*speABCDEF* Δ*potAB* Δ*potFGHI* (orange) in M9 media supplemented with putrescine (green) or spermidine (purple) as indicated. Growth was determined by measuring OD$_{660}$. *n* = 3 or 5. (E) Measurement of intracellular spermidine contents from the WT, Δ*speABCEDF*, and Δ*speABCEDF* Δ*potAB*

Δ*potFGHI* grown in M9 supplemented with spermidine at 3 mM. Values are means ± standard deviations of determinations in 3 independent experiments. (F, G) Assessment of polyamine uptake. Polyamine contents in culture supernatant of M9 supplemented with putrescine or spermidine at 3 mM, in which *S*Tm strains as indicated grew, were determined. Relative levels of polyamine are normalized by $OD_{660}$ values of bacterial culture. Data are shown as the means ± standard deviations of the results from 3 independent experiments. ns, not significant; *$P < 0.05$; **$P < 0.01$; one-way ANOVA followed by Dunnett's multiple comparisons test. The data underlying this figure can be found in S1 Data. *S*Tm, *Salmonella enterica* serovar Typhimurium; WT, wild-type.

respectively (Figs 1A and S1A). Deletion of the *potAB* genes had no effect on the expression of *sifA*, which is located downstream of *potAB* and encodes a T3SS-2 effector (S1B Fig).

All the polyamine mutants grew in rich medium (LB) at a level equivalent to that of the WT (S1C Fig). In contrast, in minimal medium (M9) the growth rate of all the mutants except for Δ*potAB* Δ*potFGHI* was significantly reduced compared to the WT (Fig 1B). Supplementation of the M9 culture medium with PUT or SPD restored the growth of Δ*speABCEDF* (Fig 1C). Strikingly, the polyamine supplementation restored the growth of Δ*speABCEDF* Δ*potAB* Δ*potFGHI* (Fig 1D), raising the possibility that *S*Tm strain SL1344 can import polyamines in the absence of both PotABCD and PotFGHI transporters, but to a lesser degree, as seen in *E. coli* PuuP and *Proteus mirabillis* PlaP [30,31]. To explore the possibility, we investigated the intracellular contents of polyamines of *S*Tm strains grown in M9 supplemented with SPD or without supplementation. In M9 without SPD, intracellular SPD was detected in the WT but not Δ*speABCEDF* or Δ*speABCEDF* Δ*potAB* Δ*potFGHI* (Fig 1E). In contrast, intracellular SPD was detected at similar levels in all tested strains grown in M9 containing SPD. These results indicate that Δ*speABCEDF* Δ*potAB* Δ*potFGHI* can import exogenous polyamines through an as-yet-unidentified transporter.

To characterize the role of PotABCD and PotFGHI in polyamine uptake in *S*Tm, we next measured polyamine concentrations in the culture supernatant of M9 supplemented with PUT or SPD. PUT levels in the culture supernatant of *S*Tm strains harboring mutations of *pot* genes were higher than those of the WT and Δ*speABCEDF* (Fig 1F). In contrast, SPD levels were higher in the culture supernatants of the Δ*speABCEDF* Δ*potAB* and Δ*speABCEDF* Δ*potAB* Δ*potFGHI* in comparison with those of the WT, Δ*speABCEDF* and Δ*speABCEDF* Δ*potFGHI* (Fig 1G). These data indicate that both PotABCD and PotFGHI transporters participate in PUT uptake, whereas SPD uptake in *S*Tm depends on PotABCD, not PotFGHI.

Taken together, these results show that the maintenance of cytoplasmic levels of polyamines in *S*Tm contributes to bacterial growth in the culture media. Furthermore, supplementation of polyamines can restore the retarded growth, indicating that the polyamine uptake systems play a crucial role in polyamine homeostasis in *S*Tm.

## Polyamine uptake by PotABCD transporter contributes to *Salmonella* pathogenesis in a mouse model

To uncover the role of polyamines in *Salmonella* infection, we established a mouse infection model for *Salmonella* gastroenteritis and systemic infection, and compared colonization levels of the polyamine mutants with those of the WT in the gut lumen (feces and cecum) and spleen. Streptomycin-pretreated C57BL/6 mice were orally challenged with an equal mixture of the WT and the polyamine mutants. On day 4 postinfection (p.i.), the bacterial loads in the feces, cecum, and spleen were analyzed, and competitive index (CI) values were determined. Bacterial loads of the Δ*speABCEDF*, Δ*potAB* Δ*potFGHI*, and Δ*speABCEDF* Δ*potAB* strains recovered from mouse feces and caecal content were equivalent to those of WT (Fig 2A). In contrast, the Δ*speABCEDF* Δ*potFGHI* and Δ*speABCEDF* Δ*potAB* Δ*potFGHI* were impaired in gut colonization compared to the WT. Notably, the colonization levels of the Δ*speABCEDF* Δ*potAB* Δ*potFGHI* were further reduced compared to those of Δ*speABCEDF* Δ*potFGHI* (Fig 2A). In the

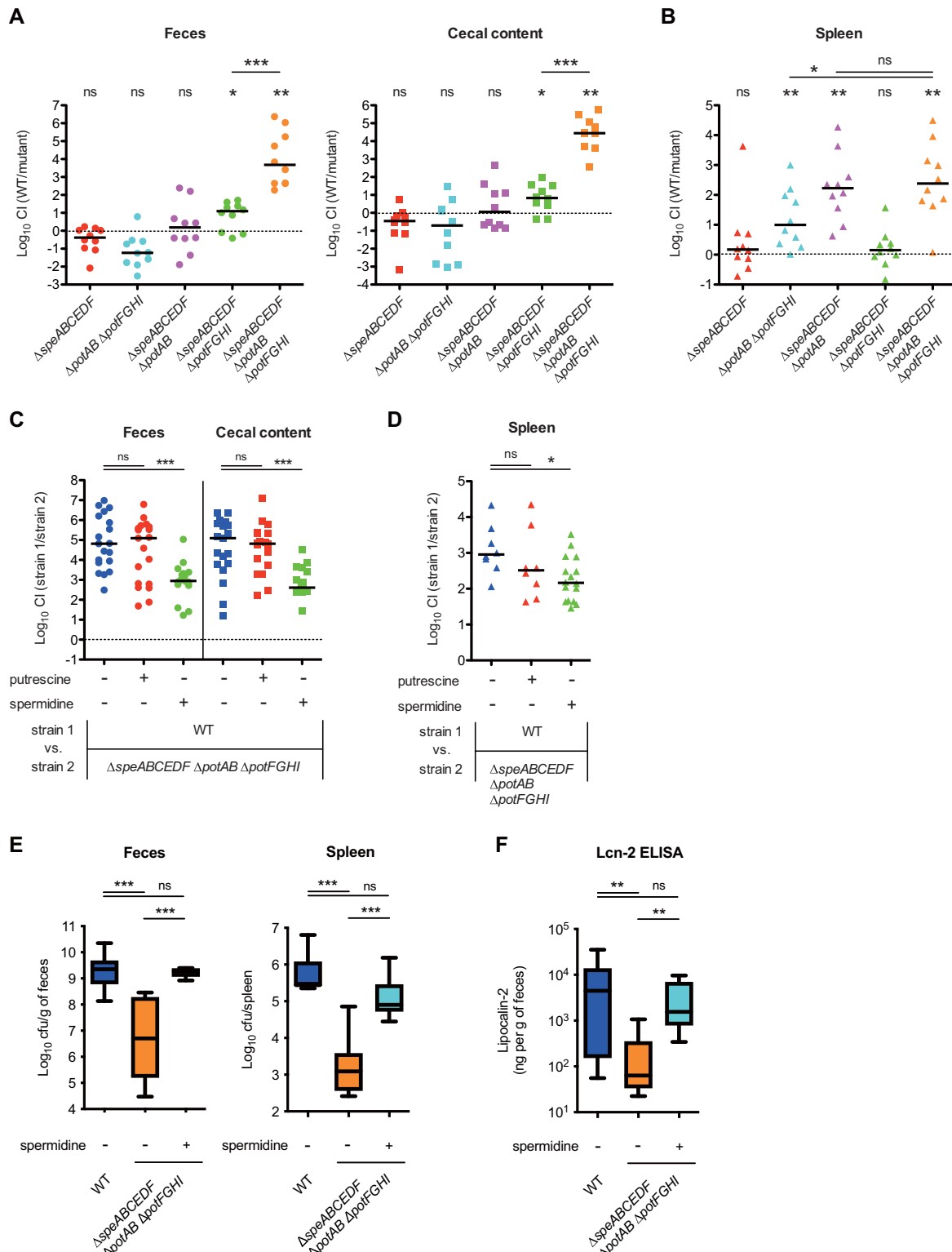

**Fig 2. Polyamine uptake contributes to *Salmonella* pathogenesis in the mouse model for gastrointestinal and systemic infection.** (A–D) The CI of *S*Tm loads from C57BL/6 on day 4 postinfection. Bars represent median values, and *n* is indicated by the number of dots. ns, not significant; \*$P < 0.05$; \*\*$P < 0.01$; \*\*\*$P < 0.001$; Wilcoxon signed-rank test and one-way ANOVA followed by Dunnett's multiple comparisons test. (E) *S*Tm loads in feces and spleen from C57BL/6 on day 4 postinfection (*n* = 8, respectively). (F) Lipocalin-2 (Lcn-2) levels in feces. The box plot with whiskers shows the ranges from minimum to maximum values, and the black bars indicate medians. ns,

not significant; \*$P < 0.05$; \*\*$P < 0.01$; \*\*\*$P < 0.001$; one-way ANOVA followed by Dunnett's multiple comparisons test. The indicated groups of mice were treated with 1% putrescine or 1% spermidine in the drinking water (C–F). The data underlying this figure can be found in S1 Data. CI, competitive index; *S*Tm, *Salmonella enterica* serovar Typhimurium.

spleen, the Δ*potAB* Δ*potFGHI*, Δ*speABCEDF* Δ*potAB*, and Δ*speABCEDF* Δ*potAB* Δ*potFGHI* had impaired colonization compared to the WT (Fig 2B). Colonization levels of the Δ*speAB-CEDF* Δ*potAB* Δ*potFGHI* were further reduced compared to that of Δ*potAB* Δ*potFGHI*, whereas they were similar to the levels of the Δ*speABCEDF* Δ*potAB*.

Introduction of a plasmid encoding *potAB* into the Δ*speABCEDF* Δ*potAB* Δ*potFGHI* tended to reverse the growth defects in feces, whereas colonization levels in the caecal content of the *potAB*-complemented mutant, but not the *potFGHI*-complemented mutant, were significantly restored in comparison with those of the control mutant strain harboring the vector control (S2A Fig). Similarly, the *potAB*-complemented mutant showed significant recovery of colonization in the spleen (S2B Fig). Furthermore, 1% SPD, but not 1% PUT, in the drinking water significantly complemented the colonization defects of the Δ*speABCEDF* Δ*potAB* Δ*potFGHI* in feces and caecal content (Fig 2C). Likewise, SPD supplementation conferred significant colonization recovery in the spleen (Fig 2D).

Next, to ask whether the reduced colonization efficiency of the polyamine mutant is involved in the induction of gut inflammation, we performed mice infection experiments with the WT or Δ*speABCEDF* Δ*potAB* Δ*potFGHI*, respectively, and evaluated gut inflammation levels by measuring the amount of intestinal lipocalin-2 (Lcn-2), an inflammatory marker. The WT, but not the Δ*speABCEDF* Δ*potAB* Δ*potFGHI*, persistently colonized the luminal gut and the spleen (Fig 2E). In contrast, addition of SPD to the drinking water reversed the colonization defects of the Δ*speABCEDF* Δ*potAB* Δ*potFGHI*. Furthermore, Lcn-2 ELISA showed that fecal levels of Lcn-2 in mice infected with the Δ*speABCEDF* Δ*potAB* Δ*potFGHI* were reduced compared to those in WT-infected mice and were restored by addition of 1% SPD to the drinking water (Fig 2F).

Collectively, these results suggest that neither polyamine homeostasis by biosynthesis nor that by transport alone has a significant effect on *S*Tm colonization in the gut and spleen. Importantly, polyamine homeostasis by uptake of exogenous SPD contributes to the *S*Tm gut colonization and induction of the intestinal inflammation and systemic spread to extraintestinal organs including the spleen, contributing to *S*Tm pathogenesis.

## Uptake-dependent polyamine homeostasis is involved in T3SS virulence in a mouse model

The 2 T3SSs play a central role in *S*Tm pathogenesis as evidenced by the results that the colonization levels of Δ*invG* Δ*ssaV* (the double-deficient genetic strain) in feces and the spleen were significantly reduced compared to WT (Fig 3A). Thus, we next investigated the contribution of the T3SSs to the polyamine homeostasis-dependent pathogenesis. The CI values from the Δ*invG* Δ*ssaV* (the 2 T3SSs-deficient genetic strain) versus the Δ*invG* Δ*ssaV* Δ*speABCEDF* Δ*potAB* Δ*potFGHI* were dramatically reduced in comparison with those from WT versus the Δ*speABCEDF* Δ*potAB* Δ*potFGHI* in the feces, cecal content, and spleen (Fig 3B and 3C), suggesting that the T3SSs are major virulence determinants in the polyamine homeostasis-dependent pathogenesis in both gastrointestinal and systemic infection.

To clarify the link between the uptake-dependent polyamine homeostasis and T3SS-1 virulence in a mouse model, we next investigated infectivity in a mouse model on day 2 p.i., in which the induction of gut inflammation depends on the T3SS-1 [32]. Streptomycin-pretreated mice were gavaged with the WT or Δ*invG* or Δ*potAB* Δ*potFGHI* or Δ*speABCEDF*

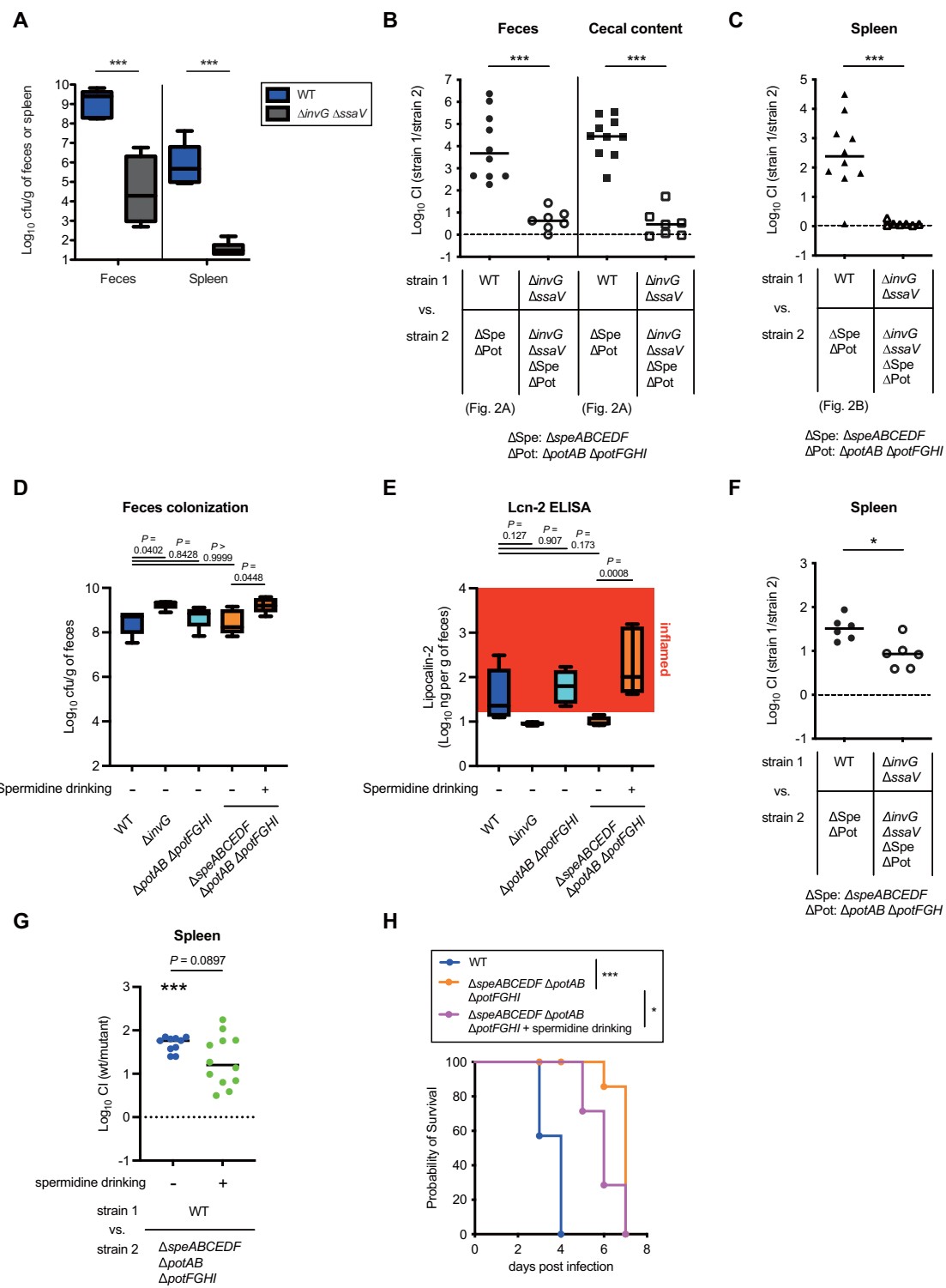

**Fig 3. T3SS-1 and T3SS-2 play a central role in the uptake-dependent *S*Tm pathogenesis.** (A) Streptomycin-pretreated C57BL/6 mice (*n* = 10, respectively) were infected by gavage with *S*Tm WT or the Δ*invG* Δ*ssaV* for 4 days. Fecal and splenic loads were determined by selective plating. (B, C) The CI of *S*Tm loads from C57BL/6 on day 4 postinfection by oral gavage. Bars represent median values, and *n* is indicated by the number of dots. ns, not significant; ***P < 0.001; Mann–Whitney U test. (D, E) Streptomycin-pretreated C57BL/6 mice (*n* = 5, respectively) were infected by gavage with the WT or Δ*invG* or Δ*potAB* Δ*potFGHI* or Δ*speABCEDF* Δ*potAB* Δ*potFGHI* or Δ*speABCEDF* Δ*potAB* Δ*potFGHI* with spermidine drinking for 2 days (*n* = 5, respectively). (D) *S*Tm loads in feces from C57BL/6 on day 2 postinfection. (E) Lcn-2 levels in feces. The box plot with whiskers shows the ranges

from minimum to maximum values, and the black bars indicate medians. One-way ANOVA followed by Dunnett's multiple comparisons test. (F, G) The CI *S*Tm loads from C57BL/6 on day 2 postinfection by intraperitoneal inoculation. Bars represent median values, and *n* is indicated by the number of dots. *$P < 0.05$; Wilcoxon signed-rank test and Mann–Whitney U test. (H) Survival of C57BL/6 mice inoculated intraperitoneally with *S*Tm WT or the Δ*speABCEDF* Δ*potAB* Δ*potFGHI* or Δ*speABCEDF* Δ*potAB* Δ*potFGHI* with spermidine drinking (*n* = 3 or 6, respectively). Survival rate (%) was calculated. *$P < 0.05$; ***$P < 0.001$; Log-rank test compared with the WT or Δ*speABCEDF* Δ*potAB* Δ*potFGHI*. The data underlying this figure can be found in S1 Data. CI, competitive index; *S*Tm, *Salmonella enterica* serovar Typhimurium; WT, wild-type.

Δ*potAB* Δ*potFGHI* for 2 days infection. On day 2 p.i., high-level gut colonization was observed in all *S*Tm strains analyzed (Fig 3D). In contrast, the gut inflammation was elicited under T3SS-1-dependent manner as evidenced by the results that fecal Lcn-2 levels were elevated in mice infected with the WT, but not the Δ*invG* (Fig 3E). It should be notable that similar to the WT, the Δ*potAB* Δ*potFGHI* induced the gut inflammation. In contrast, mice infected with the Δ*speABCEDF* Δ*potAB* Δ*potFGHI* had no gut inflammation, whereas addition of 1% SPD to the drinking water restored the reduced inflammation (Fig 3E). These results suggest that the uptake-dependent polyamine homeostasis rather than the Pot transporters is required for T3SS-1-dependent virulence in gastrointestinal infection.

T3SS-2 plays a critical role in systemic infection by contributing to the intracellular replication and induction of cell death. To bypass the T3SS-1-dependent pathogenesis, we next investigated infectivity in a mouse model by intraperitoneal infection of *S*Tm. C57BL/6 mice were infected intraperitoneally with a 1:1 mixture of the WT and Δ*invG* Δ*ssaV* or the Δ*invG* Δ*ssaV* and Δ*invG* Δ*ssaV* Δ*speABCEDF* Δ*potAB* Δ*potFGHI*. On day 2 p.i., mice were euthanized, and bacterial loads in the spleen were determined. The CI values of the Δ*invG* Δ*ssaV* veusus Δ*invG* Δ*ssaV* Δ*speABCEDF* Δ*potAB* Δ*potFGHI* were significantly reduced compared to those of WT versus the Δ*invG* Δ*ssaV* (Fig 3F). We next investigated effect of the spermidine drinking in CI experiments with the WT and Δ*speABCEDF* Δ*potAB* Δ*potFGHI*. Bacterial loads in the spleen of the Δ*speABCEDF* Δ*potAB* Δ*potFGHI* on day 2 p.i. were reduced compared to those of WT (Fig 3G). In contrast, consumption of 1% SPD in drinking water partly restored the attenuated colonization of the *speABCEDF* Δ*potAB* Δ*potFGHI*. Furthermore, mouse infection experiments for survival analysis showed that the Δ*speABCEDF* Δ*potAB* Δ*potFGHI* was impaired in pathogenesis compared to the WT (Fig 3H). The defective pathogenesis was significantly restored by supplementation of the drinking water with 1% SPD. These results suggest that the uptake-dependent polyamine homeostasis contributes to systemic infection largely involved in T3SS-2.

## Uptake-dependent polyamine homeostasis is required for functional expression of T3SS-1 by activating translocation by type 3 effector

We next examined functional activities of T3SS-1 in the Δ*speABCEDF* Δ*potAB* Δ*potFGHI*. A master transcriptional activator, HilA, regulates the expression of T3SS-1 genes such as the *sicA-sipBCDA* operon, which encode a chaperone and secreted effectors/translocators (S3A Fig). Expression levels of *sicA* measured by using a β-galactosidase assay were decreased in the Δ*hilA* mutant compared to the WT, whereas the *sicA* levels were similar between the Δ*speABCEDF* Δ*potAB* Δ*potFGHI* and WT strains (S3B Fig). Similarly, in vitro secretion of SipB and SipC was competent in both the Δ*speABCEDF* Δ*potAB* Δ*potFGHI* and WT, but not in the Δ*invG* lacking T3SS-1 secretion (S3C Fig). We next determined the ability of the polyamine mutant to invade HeLa cells. Both the Δ*speABCEDF* Δ*potAB* Δ*potFGHI* and Δ*invG* showed similar, significant reductions in their ability to invade HeLa cells (Fig 4A). Introduction of a plasmid encoding *potAB* restored the invasion capacity (S3D Fig). Similarly, addition of SPD to the bacteria growth medium completely reversed the invasion defects (Fig 4B), indicating

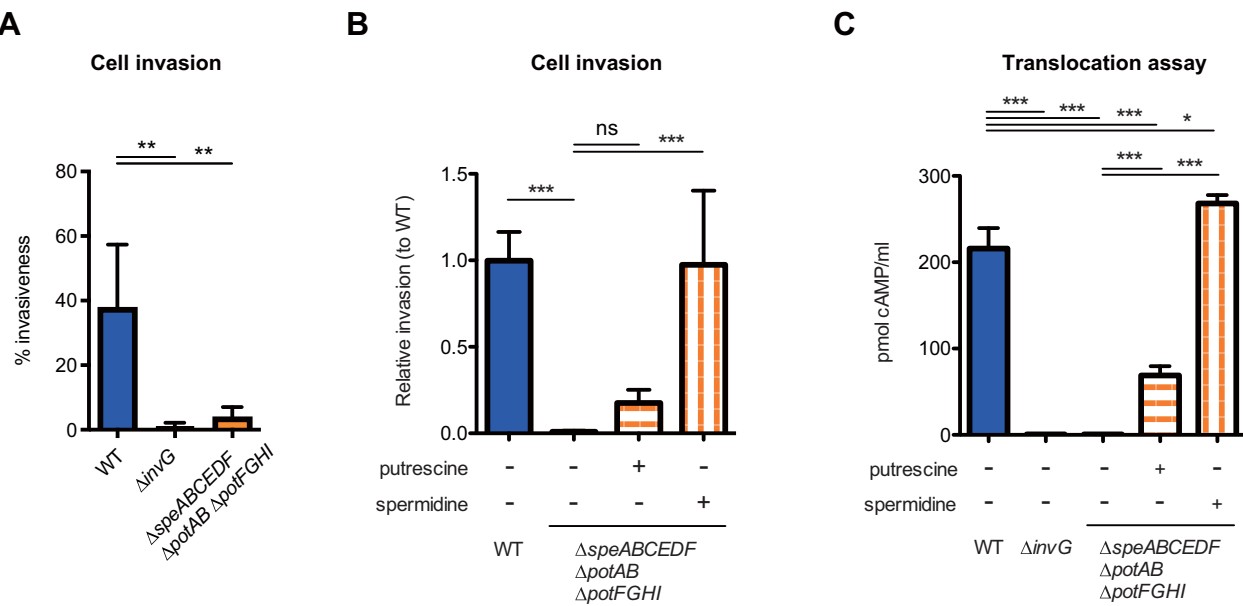

**Fig 4. Spermidine uptake is required for T3SS-1-dependent virulence by activating translocation of type 3 effector.** (A, B) HeLa cells were infected with the indicated *S*Tm strains, and the number of invading bacteria was determined by a gentamicin protection assay. The invasion rate (%) or relative invasion to WT were determined from 4 or 5 independent experiments. Bars represent means ± SD. ns, not significant; **$P < 0.01$; ***$P < 0.001$; one-way ANOVA followed by Dunnett's multiple comparisons test. (C) Translocation of SopD-CyaA fusion protein into HeLa cells in the T3SS-1-dependent fashion. Monolayers of HeLa cells were infected with the indicated *S*Tm strains expressing SopD-CyaA fusion protein for 2 h. Bars represent means ± SD from 3 independent experiments. *$P < 0.05$; ***$P < 0.001$; one-way ANOVA followed by Dunnett's multiple comparisons test. The data underlying this figure can be found in S1 Data. *S*Tm, *Salmonella enterica* serovar Typhimurium; WT, wild-type.

that the Δ*speABCEDF* Δ*potAB* Δ*potFGHI* can import SPD by unknown transporter(s), resulting in restoration of the virulence. On the other hand, the invasiveness of Δ*potAB* Δ*potFGHI* into HeLa cells was impaired (S3E Fig). The reduced invasion capacity was partly reversed by introducing a plasmid encoding *potAB* but not *potFGHI*. In contrast, unlike the Δ*speABCEDF* Δ*potAB* Δ*potFGHI*, SPD supplementation had no complementary effect on the invasion defects of the Δ*potAB* Δ*potFGHI* (S3E Fig), indicating that the PotABCD transporter contributes to the cell invasion of *S*Tm in a manner independent of polyamine uptake. Collectively, these results suggest that polyamine homeostasis by uptake of exogenous PUT and SPD is required for T3SS-1-dependent cell invasion.

To further decipher the role of PUT and SPD in T3SS-1 activities, we investigated T3SS-1-dependent host cell interactions, i.e., contact hemolysis, cell cytotoxicity, replication within host cells, and effector translocation into host cells. T3SS-1 imposes contact hemolysis on red blood cells (RBCs) by forming pores on the cell membrane [33]. The WT had hemolytic activity, whereas no hemolysis was observed in RBCs infected with the Δ*invG* and Δ*speABCEDF* Δ*potAB* Δ*potFGHI* (S3F and S3G Fig). The hemolytic activity of the Δ*speABCEDF* Δ*potAB* Δ*potFGHI* was restored by introduction of a plasmid encoding *potAB* (S3F and S3G Fig). Next, we examined T3SS-1-dependent cytotoxicity in RAW264.7 cells [34,35]. The WT-induced cell death in a T3SS-1-dependent manner based on the finding that the LDH release was decreased in the Δ*invG* (S3H Fig). Similar to the Δ*invG*, the Δ*speABCEDF* Δ*potAB* Δ*potFGHI* showed reduced cytotoxic activity, which were partially restored by addition of SPD to the bacteria growth medium. The WT replicated within RAW264.7 cells, whereas the Δ*speABCEDF* Δ*potAB* Δ*potFGHI* exhibited a survival defect, which tended to be reversed by supplementation with SPD (S3I Fig).

Previous results (S3D and S3G Fig.) have shown that overexpression of the PotABCD transporter from the plasmid pMW-*potAB* has much more than a complementary effect, raising the possibility that the PotABCD transporter facilitates bacterial growth independently of SPD uptake. Thus, we have examined the growth of the Δ*speABCEDF* Δ*potAB* Δ*potFGHI* harboring pMW-*potAB* in M9 with or without SPD. In M9 without SPD, the Δ*speABCEDF* Δ*potAB* Δ*potFGHI* harboring pMW-*potAB* exhibited retarded growth compared to the WT (S4 Fig). In contrast, SPD supplementation led to enhanced growth on the Δ*speABCEDF* Δ*potAB* Δ*potF-GHI* harboring pMW-*potAB*. These results indicate that expression of the PotABCD transporter contributes to bacterial growth through the uptake of exogenous SPD.

Finally, we investigated whether polyamine homeostasis affects the translocation of T3SS-1 effectors into host cells. A translocation assay using a CyaA reporter system with the T3SS-1 effector SopD showed that SopD was translocated into HeLa cells in a T3SS-1-dependent fashion (Fig 4C). Like the Δ*invG*, the Δ*speABCEDF* Δ*potAB* Δ*potFGHI* failed in the translocation of SopD. Addition of SPD to the bacteria growth medium significantly restored the translocation in the Δ*speABCEDF* Δ*potAB* Δ*potFGHI*, whereas partial restoration was observed by PUT supplementation (Fig 4C). SPD supplementation had no effect on the Δ*invG*, indicating that SPD-dependent enhancement of effector translocation depends on the functionality of T3SS-1 (S3J Fig). Furthermore, to exclude the possibility that invasion efficiency defines the translocation levels, we assessed the translocation in early stages of invasion. At 30-min postinfection, the number of cell-associated bacteria was similar in all tested strains (S5A Fig). In this stage, SopD was translocated into HeLa cells infected with the WT, but not the Δ*invG* (S5B Fig). Translocation levels in the Δ*speABCEDF* Δ*potAB* Δ*potFGHI* were equivalent to those in the Δ*invG*, whereas the Δ*speABCEDF* Δ*potAB* Δ*potFGHI* grown in LB supplemented with SPD showed enhanced translocation (S5B Fig). Intracellular expression levels of SopD-CyaA fusion proteins were similar in all tested strains (S5C Fig). Collectively, these results indicate that uptake of polyamines, especially SPD, plays a critical role in the functional expression of T3SS-1 in the pathogen–host cell interaction stages, which include the process such as translocation of T3SS-1 effectors.

## Spermidine uptake is required for T3SS-2-dependent virulence

To clarify the role of polyamine homeostasis in T3SS-2 activities, we first confirmed the effect of mutations of polyamine genes on the transcriptional activities of T3SS-2 genes. RNA extracted from STm strains grown in minimal medium low in phosphate and magnesium (LPM), a T3SS-2-inducible medium, was subjected to RT-qPCR analysis with primers specific to T3SS-2 genes. The expression levels of all the tested genes were reduced in the Δ*ssrB* lacking a pivotal regulator for T3SS-2 (S6A Fig). In contrast, the Δ*speABCEDF* Δ*potAB* Δ*potFGHI* showed similar *sseJ* expression, but higher-level expression of *sseB* and *ssaG*, compared to the WT. We next investigated whether the Δ*speABCEDF* Δ*potAB* Δ*potFGHI* secretes T3SS-2 substrate extracellularly. Secretion of SseB was detected in the WT and Δ*speABCEDF* Δ*potAB* Δ*potFGHI*, whereas no secretion was observed in the Δ*ssaV*, which had nonfunctional T3SS-2 machinery (S6B Fig). These results indicated that the Δ*speABCEDF* Δ*potAB* Δ*potFGHI* can secrete T3SS-2 substrate extracellularly.

We next determined the capacity of the Δ*speABCEDF* Δ*potAB* Δ*potFGHI* for replication in RAW264.7 cells. Similar to the results of invasion into HeLa cells, we found that the invasiveness of Δ*speABCEDF* Δ*potAB* Δ*potFGHI* into RAW264.7 cells was lower compared to the WT, a deficit that was restored by transformation of a plasmid encoding *potAB* (S6C Fig). Furthermore, we found that the Δ*speABCEDF* Δ*potAB* Δ*potFGHI* was impaired in replication within RAW264.7 cells, in which the Δ*ssaV* lacking T3SS-2 secretion had reduced growth (Fig 5A).

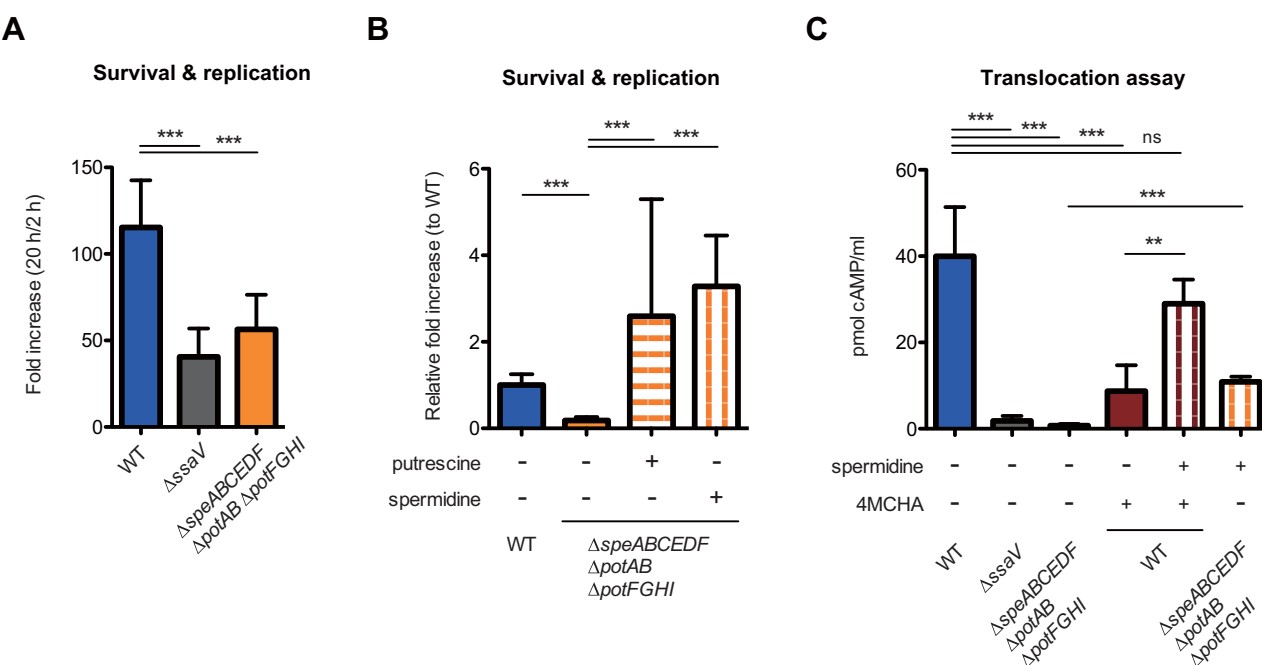

**Fig 5. Spermidine uptake is required for T3SS-2-dependent virulence by activating translocation of type 3 effector.** (A, B) Survival and replication within RAW264.7 cells. RAW264.7 cells were infected with the indicated *S*Tm strains for 20 h, and the ability to survival and replicated within RAW264.7 cells were determined as fold increase. Bars represent means ± SD from at least 5 independent experiments. *$P < 0.05$; ***$P < 0.001$; one-way ANOVA followed by Dunnett's multiple comparisons test. (C) Intracellular levels of SseJ-CyaA fusion protein, translocated in RAW264.7 cells in the T3SS-2-dependent fashion. Monolayers of RAW264.7 cells were infected with the indicated *S*Tm strains expressing SseJ-CyaA fusion protein for 16 h. Bars represent means ± SD from at least 3 independent experiments. ns, not significant; **$P < 0.01$; ***$P < 0.001$; one-way ANOVA followed by Dunnett's multiple comparisons test. The data underlying this figure can be found in S1 Data. *S*Tm, *Salmonella enterica* serovar Typhimurium.

Restored replication activities were observed in the complementation mutant strain harboring a plasmid encoding *potAB*, but not *potFGHI* (S6D Fig). Furthermore, addition of PUT or SPD into the cell culture medium restored the replication of the Δ*speABCEDF* Δ*potAB* Δ*potFGHI* (Fig 5B). These results suggest that uptake-dependent polyamine homeostasis is required for survival and replication in macrophages.

*S*Tm induces T3SS-2-dependent cell death in macrophages at the delayed stages of infection [36]. The WT induced T3SS-2-dependent cytotoxicity, which was not seen in RAW264.7 cells infected with the Δ*ssaV* (S6E Fig). In addition to the Δ*ssaV*, the Δ*speABCEDF* Δ*potAB* Δ*potFGHI* exhibited reduced cytotoxicity activity. Addition of SPD to the cell culture medium partially restored the cytotoxic activity (S6E Fig). The effect of SPD supplementation depended on T3SS-2-dependent cytotoxicity (S6F Fig). The results suggest that SPD uptake-mediated polyamine homeostasis is required for the T3SS-2-dependent cytotoxicity. Moreover, to better understand the requirement of intracellular SPD, which is synthesized and imported to *S*Tm, RAW264.7 cells were treated with trans-4-methylcyclohexylamine (4MCHA), a spermidine synthase (SRM) inhibitor, to decrease cellular SPD levels [37]. Treatment with 4MCHA significantly reduced the T3SS-2-dependent cytotoxic activity by WT (S6G Fig). The reduced cytotoxic activity was restored by adding SPD to the cell culture medium. Similar results were obtained for the T3SS-2-dependent survival and replication in RAW264.7 cells (S6H Fig).

We next examined translocation of the T3SS-2 effector into host cells by using a CyaA reporter system. Intracellular expressions of SseJ-CyaA fusion proteins were similar in all tested strains (S6I Fig). A T3SS-2 effector SseJ was translocated into RAW264.7 cells infected with the WT, whereas barely detectable levels were observed in the Δ*ssaV* and Δ*speABCEDF*

Δ*potAB* Δ*potFGHI* (Fig 5C). Treatment with 4MCHA reduced the intracellular levels of SseJ translocated from the WT, and these levels were restored by addition of SPD to the cell culture medium. Similarly, SPD supplementation partially restored defective translocation in the Δ*speABCEDF* Δ*potAB* Δ*potFGHI* (Fig 5C). Collectively, these results suggest that cell-derived SPD uptake-dependent polyamine homeostasis is required for functional expression of T3SS-2, which confers various functions onto macrophages in *S*Tm, including cell survival, cell replication, cell death, and effector translocation.

## The *S*Tm Δ*speABCEDF* Δ*potAB* Δ*potFGHI* mutant fails to assemble the needle complex of T3SS machinery

To uncover the underlying mechanism by which the Δ*speABCEDF* Δ*potAB* Δ*potFGHI* is impaired in the functional expression of T3SS, we next compared the T3SS-1 machinery between the WT and mutant. The core component of T3SS is the needle complex, which consists of basal body rings and a needle [38,39]. Thus, the T3SS-1 needle complex was prepared from Δ*fliGHI* genetic background strains lacking flagella and visualized by electron microscopy. Needle complexes of WT largely included the basal body structure with the needle, whereas needles were missing in all needle complexes prepared from the Δ*speABCEDF* Δ*potAB* Δ*potFGHI* (Fig 6A). Furthermore, we investigated the proportion of needle complexes with needles from WT, the needle-subunit-deficient mutant Δ*prgI*, and Δ*speABCEDF* Δ*potAB* Δ*potFGHI* grown in LB supplemented with SPD or without it. Approximately 46% of purified needle complexes from WT grown in LB had the needle (S7A and S7B Fig). In contrast, the Δ*prgI* and Δ*speABCEDF* Δ*potAB* Δ*potFGHI* grown in LB had no needles (S7A, S7D and S7F Fig). The addition of SPD to the culture medium had no effect on needle assembly in any of the tested strains (S7A,S7C, S7E and S7G Fig).

We next investigated whether PrgI, a needle component, was exported to the cell-surface in the Δ*speABCEDF* Δ*potAB* Δ*potFGHI*. Deletion of *invJ* leads to the formation of an extremely long needle under the T3SS-1-inducible condition, resulting in bacterial clumping of the culture, in which the very long needle becomes tangled [40]. *S*Tm strains used in this assay were rendered nonmotile by deletions of the *fliGHI* genes for facilitation of the clumping and were transformed with a plasmid expressing HilD, a transcriptional activator for T3SS-1 genes to induce the expression of T3SS-1 [41]. We found that the bacterial clumping occurred in the culture of Δ*invJ* under the T3SS-1-inducible condition (Fig 6B and 6C). In the Δ*speABCEDF* Δ*potAB* Δ*potFGHI*, only the spontaneous sedimentation of bacteria was observed even when L-arabinose was added to ensure the T3SS-1-inducible condition, indicating that the clumping by PrgI entanglement was canceled in the culture of the mutant strain. These results suggest that uptake-dependent polyamine homeostasis is required for the needle assembly of T3SS-1. Importantly, these results likely explain why the Δ*speABCEDF* Δ*potAB* Δ*potFGHI* is unable to insert the T3SS translocators on the host cell membrane as seen in the hemolysis assay (S3F and S3G Fig), but can secrete them extracellularly (S3C Fig).

We next investigated the expression levels of the T3SS needle subunit in the Δ*speABCEDF* Δ*potAB* Δ*potFGHI*. Initially, gene expression levels were measured by using a β-galactosidase assay. The HilA positively regulates the *prgHIJK-orgABC* operon, including the *prgI* encoding the needle subunit of T3SS-1 (S8A Fig). Expression levels of *prgH* were decreased in the Δ*hilA* compared to the WT, whereas the *prgH* levels in the Δ*speABCEDF* Δ*potAB* Δ*potFGHI* were similar to those in the WT (S8B Fig). Thus, we next examined the protein expression of the needle subunit. To this end, a plasmid expressing an HA-tagged T3SS protein was constructed and then transformed into the *S*Tm strains. Expression of PrgH-2HA and PrgI-2HA from the plasmids depended on the presence of *hilA*, whereas the protein expression levels of both

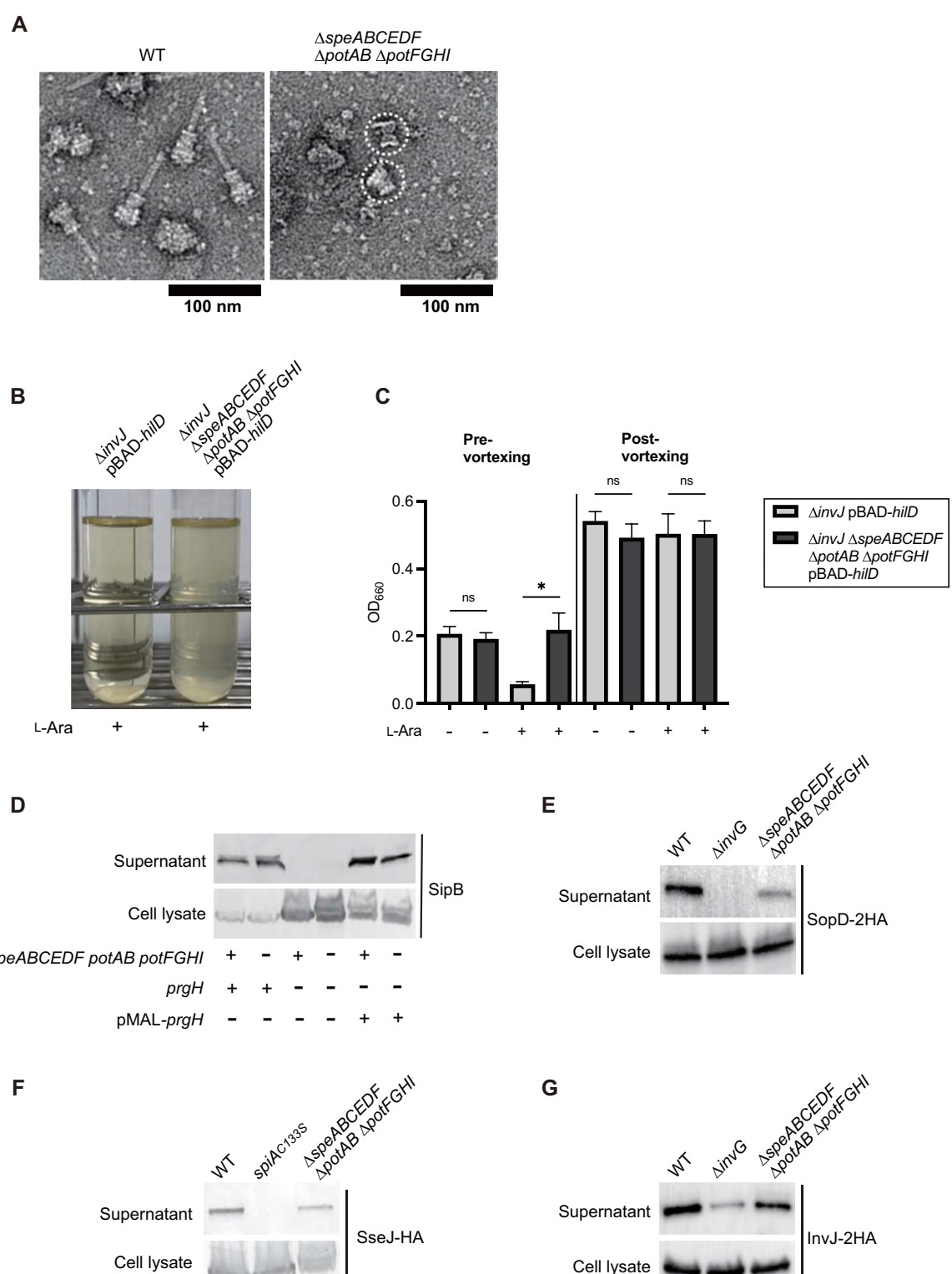

**Fig 6. Uptake-dependent polyamine homeostasis is required for assembly of needle of T3SS machinery.** (A) Electron micrographs of T3SS-1 isolated from the *S*Tm WT (Δ*fliGHI*) and Δ*speABCEDF* Δ*potAB* Δ*potFGHI* (Δ*fliGHI*). The T3SS consists of basal body rings and a needle. Dashed circle indicates basal body rings without needle attachment. The background strain used in this experiment harbored *fliGHI* deletions to prevent contamination with flagella. (B) Representative images of bacterial clumping in cultures of Δ*invJ* and Δ*invJ* Δ*speABCEDF* Δ*potAB* Δ*potFGHI*, both strains have Δ*fliGHI* genetic background and a plasmid pBAD-*hilD* expressing His-HilD to

ensure the T3SS-1 PrgI-dependent clumping. (C) Bacterial clumping was analyzed by comparing the $OD_{660}$ values pre- and post-vortexing of bacterial cultures incubated along with standing. Bars represent mean ± SD from 4 independent experiments. ns, not significant; **$P < 0.01$; ***$P < 0.001$; Student *t* test. The data underlying this figure can be found in S1 Data. (D–G) Secretion of the middle substrate (SipB), the late substrates (SopD and SseJ), and the early substrate (InvJ) from the indicated *S*Tm strains was assessed by western blot analysis with anti-SipB antiserum and anti-HA antibody. *S*Tm, *Salmonella enterica* serovar Typhimurium; WT, wild-type.

proteins in the Δ*speABCEDF* Δ*potAB* Δ*potFGHI* were equivalent to those in the WT (S8C and S8D Fig). Likewise, the protein expression levels of SsaG-2HA, the needle subunit of T3SS-2, were reduced in the Δ*ssrB*, whereas the Δ*speABCEDF* Δ*potAB* Δ*potFGHI* produced the SsaG-2HA at the similar levels to the WT (S8E Fig). These results suggest that the incomplete needle complex of T3SS machinery found in the Δ*speABCEDF* Δ*potAB* Δ*potFGHI* is not attributed to decreased expression of the needle subunit, but rather raises the possibility of impaired assembly of the needle subunit.

Our results showed that SipB is secreted extracellularly in vitro in the Δ*speABCEDF* Δ*potAB* Δ*potFGHI* (S3C Fig), indicating that the secretion switch is shifted from the early substrate to the middle substrate. Furthermore, it is noted that in general *S*Tm strains lacking the T3SS-1 needle are unable to secrete the middle and late substrates including translocators and effectors extracellularly [42]. Thus, we next asked whether secretion of the middle substrate SipB, a T3SS-1 translocator, in the Δ*speABCEDF* Δ*potAB* Δ*potFGHI* depends on T3SS-1. To this end, we introduced a mutation of *prgH* encoding a component of the T3SS-1 basal body to *S*Tm Δ*fliGHI* genetic background strains and analyzed the T3SS-1-dependent secretion of SipB. Secretion of SipB was observed in the WT and Δ*speABCEDF* Δ*potAB* Δ*potFGHI*, whereas introduction of a mutation of *prgH* abolished the SipB secretion, which restored by transformation with a plasmid expressing PrgH proteins (Fig 6D), indicating that SipB is secreted through the T3SS-1 machinery lacking the needle in the Δ*speABCEDF* Δ*potAB* Δ*potFGHI*.

We next asked whether a late substrate is secreted extracellularly in the Δ*speABCEDF* Δ*potAB* Δ*potFGHI*. To this end, secretion assays were conducted by transforming *S*Tm strains with a plasmid expressing an HA-tagged effector of T3SS-1 or -2. Furthermore, to detect the extracellular T3SS-2 effector, we employed a strain from an *ssaL*-deleted genetic background, which confers enhanced secretion of T3SS-2 effectors in vitro [43]. Notably, although the secretion levels of SopD-2HA were reduced in the Δ*speABCEDF* Δ*potAB* Δ*potFGHI* compared to the WT, the mutant could secrete the T3SS-1 effector (Fig 6E). Similar results were obtained in T3SS-2, in which secretion of SseJ-HA was detected in the WT and Δ*speABCEDF* Δ*potAB* Δ*potFGHI*, whereas no secretion was observed in the *spiA*$_{C133S}$ having nonfunctional T3SS-2 machinery [44] (Fig 6F).

InvJ, an early substrate, elicits the substrate switch to middle substrate by assisting in the localization of the inner rod protein PrgJ in the needle complex [40]. The complete T3SS machine does not contain InvJ, due to its release into the culture through the immature needle complex [45]. Thus, we finally asked whether InvJ works as a substrate switch in the Δ*speABCEDF* Δ*potAB* Δ*potFGHI*. To this end, the amount of InvJ released into the culture was compared. We found that InvJ was secreted extracellularly via the T3SS-1 machinery without a needle in the Δ*speABCEDF* Δ*potAB* Δ*potFGHI* (Fig 6G), indicating that an InvJ-dependent substrate switch may function in the Δ*speABCEDF* Δ*potAB* Δ*potFGHI*, leading to secretion of the middle and late substrates. Based on these results, it is reasonable to expect that the substrate switch in the Δ*speABCEDF* Δ*potAB* Δ*potFGHI* functions aberrantly, and confers the secretion of T3SS substrates via the machinery without the needle on this mutant. Notably, the needle assembly defect and aberrant substrate switch seen in the Δ*speABCEDF* Δ*potAB* Δ*potFGHI* could be true in the T3SS-2.

To clarify the requirement of needle assembly for secretion of T3SS substrates, we asked whether the Δ*prgI*, which lacks the needle subunit, can secrete a middle substrate (SipB) and a late substrate (SopD) into the culture media. The WT and Δ*speABCEDF* Δ*potAB* Δ*potFGHI* secreted SipB into the culture media at the similar levels (S9A Fig). In contrast, the secretion of SipB was not observed in the Δ*invG* and Δ*prgI*. Similar results were observed in the secretion of SopD-2HA (S9B Fig). Furthermore, we confirmed that the Δ*prgI* failed the substrate translocation (SopD-CyaA) and invasion into HeLa cells (S9C and S9D Fig). These results indicate that assembly of the needle subunits is required for secretion of the T3SS substrates. Furthermore, the results raise the possibility that the Δ*speABCEDF* Δ*potAB* Δ*potFGHI* can transiently express the needle for the secretion of T3SS substrates.

## Decline in host polyamine levels reduces *S*Tm colonization in the spleens of the model mice

To investigate whether reduced polyamine levels in the host are involved in the infectivity of *Salmonella*, we used difluoromethylornithine (DFMO) [46], the polyamine biosynthetic enzyme inhibitor, to control the levels of host polyamines in mouse infection experiments. Addition of 1% DFMO to drinking water significantly reduced PUT and SPD contents in the spleen, but not in feces, whereas the levels of SPM in the spleen were not reduced (Fig 7A and 7B). Next, the DFMO-supplemented mice were subjected to the mouse model for *S*Tm infection. The *S*Tm WT colonized the luminal gut at high levels in both mice treated with DFMO and untreated mice, whereas *S*Tm loads in the spleen recovered from mice treated with DFMO were lower than those recovered from untreated mice (Fig 7C). In addition, DFMO treatment attenuated replication of the *S*Tm WT in RAW264.7 cells, but not that of the Δ*speABCEDF* Δ*potAB* Δ*potFGHI* (Fig 7D). DFMO treatment drastically reduced the levels of PUT and SPD, but not SPM, in RAW264.7 cells (S10 Fig). These results suggest that a reduction in the levels of polyamines such as PUT and SPD in host cells decreases the infectivity of *S*Tm by attenuating the intracellular replication.

## *S*Tm infection elevates polyamine levels by up-regulating the expression of arginase, leading to enhanced infectivity

In mammalian cells, L-arginine transported by solute carrier family 7 (Slc7) is a substrate for NOS2 or arginase (Arg-1 and Arg-2), resulting in production of nitric oxide (NO) and citrulline or urea and L-ornithine, respectively (Fig 8A). L-Ornithine is converted to PUT by ornithine decarboxylase (ODC), and spermidine synthase (SRM) subsequently synthesizes SPD from PUT. SPM is synthesized from SPD, and spermine oxidase (SMOX) directly catalyzes this conversion. Our above-described findings indicate that *S*Tm exploits polyamines in the intestinal lumen and other cells of its hosts in order to infect them. Therefore, it is reasonable to hypothesize that *S*Tm infection increases polyamine contents in the host, resulting in enhanced infection efficiency. To examine this possibility, we compared the levels of polyamines recovery between mice infected with *S*Tm and uninfected mice. The results showed that the fecal levels of PUT and SPD in mice infected with *S*Tm WT were elevated compared to those in uninfected mice (Fig 8B). Similarly, the levels of the polyamines PUT, SPD, and SPM in the spleen of the infected mice were higher than those of uninfected mice (Fig 8C).

We next asked whether the elevation of polyamine levels in the mice infected with *S*Tm is part of the host inflammatory response. To this end, polyamine contents in the feces and spleen from a mouse model of dextran sulfate sodium (DSS)-induced colitis were measured. As expected, the DSS model mouse showed higher polyamine levels in feces and spleen

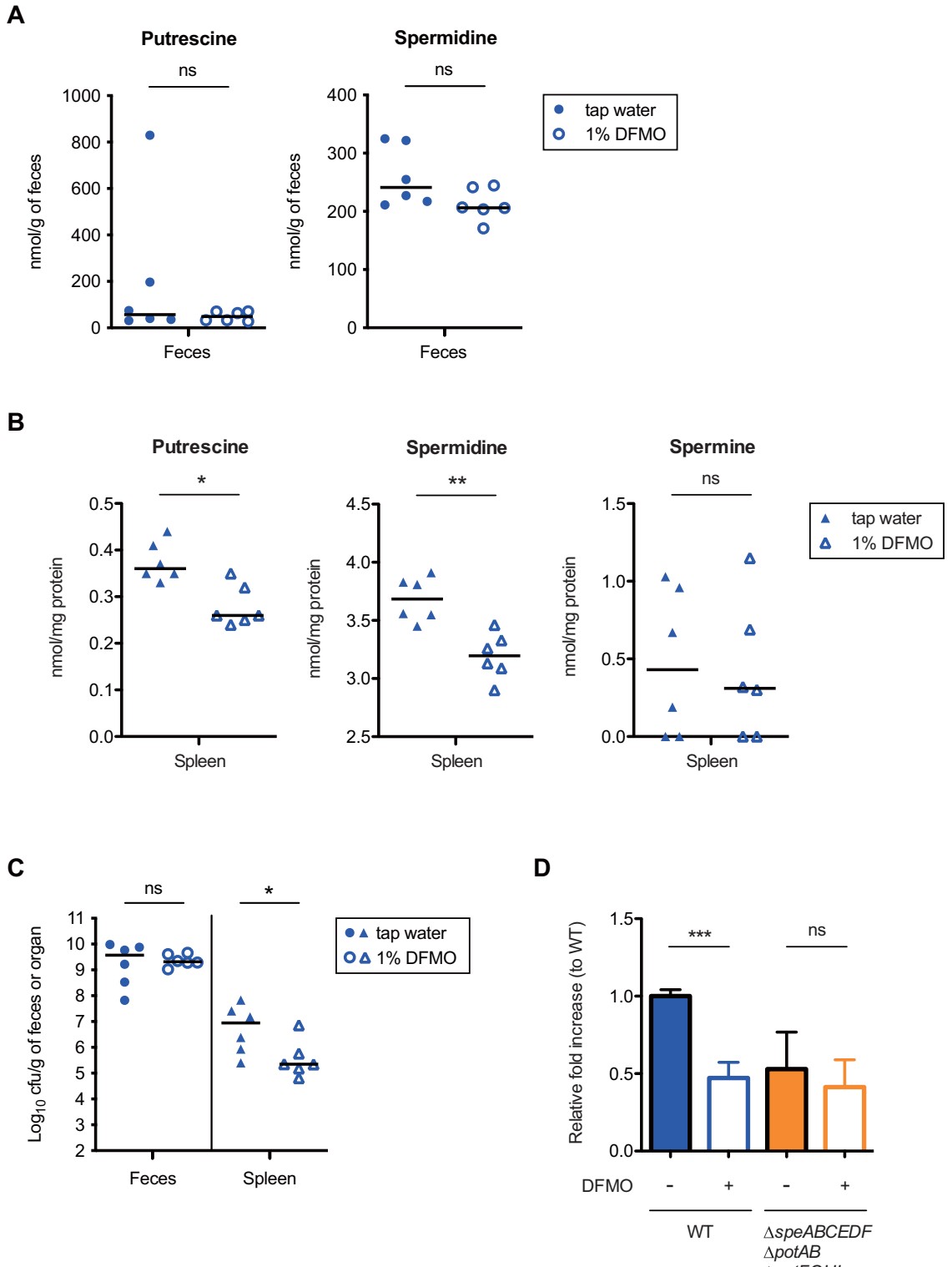

**Fig 7. Declining levels of polyamines by DFMO supplementation attenuates *Salmonella* pathogenesis.** (A, B) Polyamine levels of feces and spleen from mice supplemented with 1% DFMO in the drinking water. (C) *S*Tm WT loads in feces and spleen from the infected mice supplemented with 1% DFMO in the drinking water. Bars represent median values, and *n* is indicated by the number of dots. ns, not significant; *$P < 0.05$; **$P < 0.01$; Mann–Whitney U test. (D) Survival and replication within RAW264.7 cells. RAW264.7 cells were infected with the indicated *S*Tm strains for 20 h, and the ability to survival and replicated within RAW264.7

cells were determined as relative fold increase to WT. As indicated, 1 mM DFMO was added to cell culture media. Bars represent means ± SD from at least 3 independent experiments. ns, not significant; ***$P < 0.001$; Student *t* test. The data underlying this figure can be found in S1 Data. DFMO, difluoromethylornithine; *S*Tm, *Salmonella enterica* serovar Typhimurium; WT, wild-type.

compared to the untreated mouse (S11A and S11B Fig), lending support to the hypothesis that host inflammation facilitates the production of polyamines.

To decipher the underlying molecular mechanism by which *S*Tm induced host polyamine production, we investigated the expression levels of genes involved in polyamine biosynthesis in host cells. Thus, we measured the expression levels of the genes encoding the transporters and enzymes involved in polyamine metabolism in the spleens recovered from mice infected with the *S*Tm WT or Δ*invG* Δ*ssaV* or with PBS as a control. Expression levels of the gene encoding tumor necrosis factor α (TNF-α), an inflammation mediator, were increased in the spleens of mice infected with the WT, but not in the spleens of mice infected with the Δ*invG* Δ*ssaV* or the PBS-injected controls, indicating that *S*Tm infection leads to increased expression of specific gene such as *Tnfa* in a virulence factor-dependent fashion (Fig 8D). Expression levels of *Slc7a1* and *Slc7a2* were not increased in either mice infected with the WT or those infected with the Δ*invG* Δ*ssaV* (Fig 8E). *NOS2* expression was not increased, whereas *S*Tm WT infection significantly increased the expression levels of *Arg-1* and *Arg-2*. The induction of *Arg* gene expression was not observed in infection with Δ*invG* Δ*ssaV* (Fig 8E). Furthermore, *ODC* and *SRM* expressions in mice infected with WT were slightly induced compared to the control. In contrast, the expression levels of *SMOX* were not increased in WT-infected mice. Arginase activities in the spleens recovered from mice infected with the WT were higher than those in the spleens of mice infected with Δ*invG* Δ*ssaV* or uninfected mice (S12A Fig). Furthermore, the abundance of L-ornithine, a substrate for polyamine synthesis, in the spleens of mice infected with the WT was decreased compared to that in uninfected mice, whereas NOS2-produced L-citrulline content was equivalent between the infected and uninfected mice (S12B Fig), indicating that arginase-induced L-ornithine was rapidly consumed to produce polyamines, and the lack of change in *NOS2* expression did not affect the abundance of citrulline. These results raised the possibility that the increased levels of polyamines in the *S*Tm-infected mice were attributable to the induced expression of arginase, which could have enhanced the consumption of L-ornithine. To a lesser degree, the induction of *ODC* and *SRM* expression might contribute to the enhanced production of polyamines in host cells.

To investigate a direct link between SPD content in the host and *S*Tm infection, we infected mice given 1% SPD in drinking water with the *S*Tm WT and compared the bacterial loads in their spleens with those of *S*Tm WT-infected mice given water without SPD. SPD supplementation in drinking water led to increased levels of SPD, but not PUT, in feces (S13A Fig). Similar elevation of SPD was observed in the spleen, whereas PUT levels were significantly reduced, possibly due to antizyme activity [47] (S13B Fig). On day 4 postinfection, fecal colonization was similar in both groups of mice, whereas *S*Tm loads in the spleens of SPD-supplemented mice were significantly higher than those in mice without SPD supplementation (Fig 8F). Likewise, survival curves for the group of *S*Tm-infected mice showed that supplementation with SPD enhanced the virulence of *S*Tm, leading to earlier death of the mice (Fig 8G). These results clearly indicate a possible link between increased SPD levels and enhanced *Salmonella* pathogenesis.

## Discussion

A growing body of evidence indicates that polyamines are involved in numerous biological processes and thereby influence not only physiological actions but also the virulence

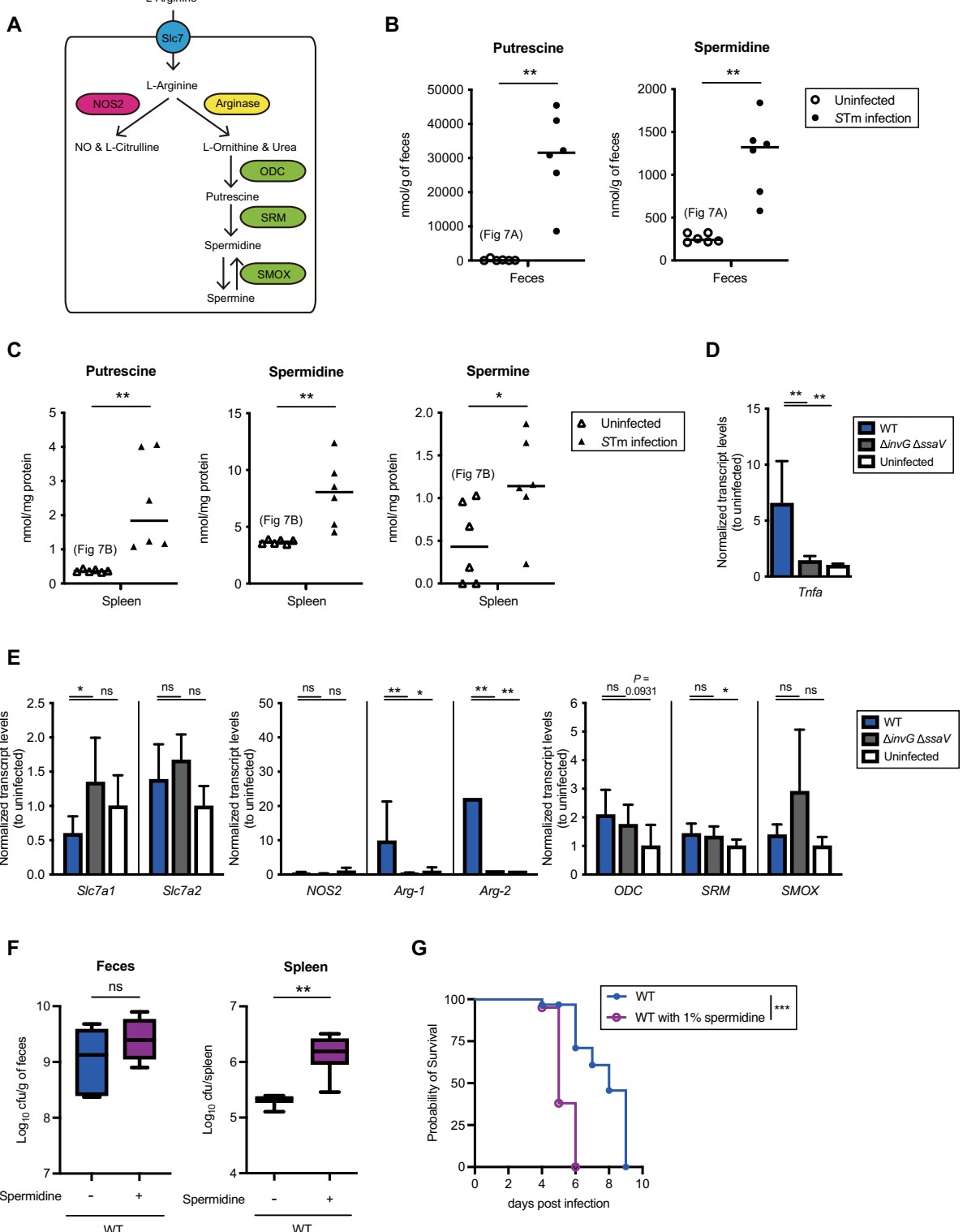

**Fig 8. *Salmonella* infection elevates host polyamine levels, resulting in enhanced virulence.** (A) L-arginine metabolic pathway in mammalian cell. (B, C) Polyamine levels of feces and spleen from STm-infected mice or uninfected mice. Bars represent median values, and *n* is indicated by the number of dots. *$P < 0.05$; **$P < 0.01$; Mann–Whitney U test. (D, E) Streptomycin-pretreated C57BL/6 mice were infected by gavage with STm WT or Δ*invG* Δ*ssaV* for 4 days. Transcripts of the indicated murine genes in spleen are presented as relative amounts to uninfected mice. Bars represent means ± SD (*n* = 6). ns, not significant; *$P < 0.05$; **$P < 0.01$; one-way ANOVA followed by

Dunnett's multiple comparisons test. (F) *S*Tm loads in feces and spleen from C57BL/6 on day 4 postinfection. The box plot with whiskers shows the ranges from minimum to maximum values, and the black bars indicate medians ($n = 6$, respectively). ns, not significant; **$P < 0.01$; Mann–Whitney U test. (G) Survival of C57BL/6 mice inoculated by gavage with *S*Tm WT ($n = 8$ or 10). Survival rate (%) was calculated. ***$P < 0.001$; Log-rank test compared with WT. The indicated groups of mice were treated with 1% spermidine in the drinking water (F, G). The data underlying this figure can be found in S1 Data. *S*Tm, *Salmonella enterica* serovar Typhimurium; WT, wild-type.

phenotypes of bacterial pathogens. More recently, it has been shown that *Salmonella* utilizes SPD for the expression of the genes involved in the giant adhesin SiiE, flagella, and T3SS-1, contributing to the early stages of pathogenesis [29]. Our findings show that polyamine uptake, especially uptake of SPD by *Salmonella*, plays a critical role in the functional expression of virulence determinant such as the T3SS. Therefore, an *S*Tm strain lacking genes associated with polyamine synthesis and uptake (Δ*speABCEDF* Δ*potAB* Δ*potFGHI*) exhibited impaired infectivity ability in the mouse model of gastrointestinal and systemic infection. *S*Tm infection led to increased levels of host polyamines by activating arginase expression. In addition, SPD addition to the drinking water of mice promoted *Salmonella* pathogenesis. Importantly, the Δ*speABCEDF* Δ*potAB* Δ*potFGHI* was unable to assemble the needle of T3SS machinery, indicating that the assembly of T3SS machinery is required for uptake-dependent polyamine homeostasis of *S*Tm. Since T3SS is the central virulence determinant in *Salmonella* pathogenesis, our findings indicate that the arginase–polyamine immunometabolism signalling axis is a critical driver of *S*Tm infection.

Polyamine has a critical role in *S*Tm virulence, as evidenced by the finding that uptake of polyamines, particularly SPD, is required for the functional expression of T3SS. An earlier work showed that polyamine positively regulates T3SS genes [25]. This result would appear to be in conflict with our present finding that polyamine levels were unrelated to the expression of T3SS genes. The disparity might be attributable to the difference in growth conditions—namely, in this study we analyzed *S*Tm cells grown to late logarithmic growth phase in LB containing 0.3 M NaCl (for T3SS-1) or grown overnight in LPM (for T3SS-2), in which 2 distinct T3SSs were maximally activated, respectively, whereas the earlier work employed *S*Tm cells grown overnight in M9 media. Furthermore, our results showed that the Δ*speABCEDF* Δ*potAB* Δ*potFGHI* expresses translocator and effector proteins intracellularly and secretes them extracellularly, indicating that this mutant can build T3SS-1 and -2 machineries that are competent for extracellular secretion of translocators and effectors. Nevertheless, effector translocation was not detected in cells infected with the Δ*speABCEDF* Δ*potAB* Δ*potFGHI* or in the T3SS mutant strain that lacked the ability to build T3SS machinery. We speculate that this was due to the incomplete pore formation on host cells by T3SS injectisome, as evidenced by the fact that the Δ*speABCEDF* Δ*potAB* Δ*potFGHI* could not induce T3SS-dependent hemolysis through contact with RBCs. The data allowed us to imagine that the Δ*speABCEDF* Δ*potAB* Δ*potFGHI* can build an immature T3SS machinery in which effectors are secreted extracellularly, but not translocated into host cells. Finally, a comparison of the purified T3SS machineries showed that the T3SS needle complex prepared from the Δ*speABCEDF* Δ*potAB* Δ*potFGHI* was lacking the needle. It should be noted that the Δ*speABCEDF* Δ*potAB* Δ*potFGHI* is able to secrete the middle (translocator) and late substrates (effector) extracellularly in a T3SS-dependent manner, albeit without the needle of T3SS machinery. This would seem to be supported by our results that InvJ functions as a substrate switch based on its secretion in the Δ*speABCEDF* Δ*potAB* Δ*potFGHI*. Importantly, we conclude that the substrate switch is aberrant in the Δ*speABCEDF* Δ*potAB* Δ*potFGHI* because it has been shown that completion of the needle assembly allows for the substrate switch from the early to middle and late substrate [40,45]. To our best knowledge, this is the first example of the secretion of the middle and late substrates

through T3SS machinery completely lacking the needle. Moreover, our data provide the first evidence that polyamine is required for the needle assembly of T3SS.

Given our data that the Δ*speABCEDF* Δ*potAB* Δ*potFGHI* can secrete T3SS effectors into culture media, it might be reasonable to speculate that the needle is transiently assembled in this mutant because the needle-subunit-deficient mutant Δ*prgI* cannot secrete any T3SS effectors. On the other hand, we further speculate that the needle assembly is unstable in the Δ*speABCEDF* Δ*potAB* Δ*potFGHI* for any reason. In this model, the assembled needle easily falls out of the basal body of T3SS. In contrast, our results showed that addition of SPD into the culture media restore the translocation of T3SS effector in the Δ*speABCEDF* Δ*potAB* Δ*potFGHI*. Therefore, the SPD supplementation may transiently facilitate assembly of the needle, but was not sufficient for stable expression of the needle, as shown by the results that addition of SPD into the culture media does not restore the needle assembly in the Δ*speABCEDF* Δ*potAB* Δ*potFGHI*.

How may polyamines facilitate needle assembly? Polyamines are ubiquitous in nature and mammalian cells at high levels (greater than millimolar concentrations), and it is notable that polyamines can interact with proteins by serving as counterions to anionic targets. Indeed, SPD and SPM play regulatory roles in microtubule assembly in cells by controlling the transition of microtubule structures between straight and curved, leading to the formation of bundles [48,49]. Moreover, it is shown that polyamines control the assembly of neuronal receptors while also acting a gatekeeper [50–53]. Based on the previous observations, it is reasonable to speculate that SPD may interact with the needle subunits through ionic binding, thereby facilitating the assembly. Moreover, the low isoelectronic point values of the needle subunits, PrgI and SsaG, might facilitate the interaction with polyamines that possess a cationic nature. Furthermore, one may consider an alternative scenario: "Indirect involvement." The Δ*speABCEDF* Δ*potAB* Δ*potFGHI* has pleiotropic effects [23,25,29]. Thus, in this mutant, the needle is assembled, but may fall out of the basal body during bacterial growth. Namely, spermidine might interact with as-yet-unidentified proteins, leading to the needle assembly. How polyamines contribute to the assembly of the needle complex is a fascinating subject for future studies.

L-arginine is a decisive factor for determining subtypes of macrophage because this amino acid is a common substrate for both NOS2 and arginase, and their respective products exert opposite effects. NOS2-activated classically activated macrophages (also known as M1 [54]) have high levels of pro-inflammatory cytokines such as TNF-α and IL-6, and NOS2 producing RNS that limit the survival of *S*Tm [55,56]. By contrast, alternatively activated macrophages (also known as M2 [54,57]) are characterized by exhibiting the arginase activation and L-ornithine production, under which *S*Tm preferentially replicates [56,58–60]. NOS2-producing RNS fulfills microbicidal functions by damaging lipids, proteins, and DNA, whereas polyamines and the L-proline produced from L-ornithine catabolism execute anti-inflammatory functions such as cell division and collagen synthesis, probably in order to support pathogen replication. In line with an earlier report [26], we found that the expressions of both types of arginase, Arg-1 and Arg-2 [61,62], in the murine spleen were increased by *S*Tm infection. It is noted that the activation of arginase depends upon the T3SS, as evidenced by our results that arginases were not increased by infection with an avirulent *S*Tm strain that are defective for both T3SS-1 and -2. Furthermore, our data showed that polyamine levels in the murine luminal gut and spleen were increased upon *S*Tm infection, which was attributed to an elevation of arginase expression. Earlier studies reported a positive correlation between host polyamine production and infection by other pathogens [63–67]. Thus, it is reasonable to hypothesize a common infection strategy by which the arginase–polyamine immunometabolism signalling axis yields 2 beneficial effects for the pathogen: the creation of a protective niche (M2

macrophage), in which the pathogens evade immune responses such as antimicrobial agents and replicate, and the production of molecules (polyamines) required for functional expression of virulence factors. In consideration of all the above, we propose an infection loop: in the gut *S*Tm imports the induced polyamines and thereby expresses the functional T3SS-1, resulting in invasion into intestinal epithelial cells and induction of gut inflammation. Subsequently, in the later stages of infection, the polyamine uptake by *S*Tm leads to the functional expression of T3SS-2 allowing for replication within macrophages, in which the translocated SteE effector via T3SS-2 maintains M2 macrophages [68,69], along with the production of polyamines, which are again imported into *S*Tm (S14 Fig).

Our finding that SPD supplementation via drinking water augmented the pathogenic effects of *S*Tm in a mouse infection model demonstrates that host SPD levels influence *Salmonella* infectivity. According to our data, it is reasonable to assume that the increased virulence can be attributed to the exploitation of host polyamines by *S*Tm for its own infection. In contrast, supplementation of polyamines has direct effects on the host because polyamines have many protective actions, including antioxidant activity [70], cytokine suppression [71], stabilization and protection of DNA [72], and autophagy stimulation [73]. Indeed, polyamines have been reported to play protective roles in the host, such as for longevity in mice [74] and, in humans and other animal models, improvement of cognitive ability [75], amelioration of cardiovascular diseases and cancer [76], protection from liver fibrosis and hepatocarcinogenesis [77,78], and treatment of obesity [79]. Importantly, polyamine metabolism in the host plays a critical role in the consequences of bacterial infections, including by *Helicobacter pylori* and *Citrobacter rodentium* [63,67,71,80]. With respect to SPD, SPD-dependent modification of the eukaryotic translation initiation factor 5A protects against infection with *H*. *pylori* and *C*. *rodentium* by enhancing antimicrobial response and autophagy [80]. In addition, SPD converted from microbiota-produced PUT in host cells ameliorates DSS-induced colitis by promoting cell renewal and M1 macrophage development [81]. Thus, although SPD supplementation is likely to play a protective role against bacterial infections, our findings suggest an opposite role as well—i.e., polyamines contribute to infection with certain pathogens such as *Salmonella*. Thus, *S*Tm strategically makes the host produce polyamines and co-opts them for its own successful infection.

Based on our results, it is reasonable to assume that inhibiting polyamine import into the vacuole containing *Salmonella* (referred to as SCV, *Salmonella*-containing vacuole) might be a promising therapeutic target. To date, several putative mammalian polyamine transporters have been identified [82–88]. However, a comprehensive understanding of the process is still lacking. Therefore, a study that sheds light on the molecular mechanism by which polyamines are delivered into the SCV will be promising and necessary.

We here found that DFMO supplementation in the drinking water weakens *Salmonella* pathogenesis in a mouse model. This was attributed to reduced levels of PUT and SPD in the murine spleen. DFMO, also called eflornithine, has been approved by the US Food and Drug Administration (FDA) and the European Medicines Agency (EMA). In addition to inhibition of ODC activities, DFMO has been shown to be a partial antagonist of arginase [89,90], leading to a shift from M2 macrophages to M1 macrophages. Indeed, earlier work demonstrated that the deficiency of macrophage-derived ODC enhances M1 macrophage activation [71], indicating that DFMO leads to M1 activation along with enhanced production of nitric oxide and thereby suppresses *S*Tm infection. Therefore, our findings might provide an opportunity for extending the application of FDA- and EMA-approved DFMO to *S*Tm infection. In addition to DFMO, other means of limiting the host polyamine levels, such as a low polyamine diet or manipulating the microbiota to produce smaller amounts of polyamines, might be new therapeutic interventions for patients infected with bacterial pathogens.

## Materials and methods

### Ethics statement

All animal experiments were reviewed and approved by the Kitasato University Institutional Animal Care and Use Committee (Permit Number: 23–22). Both, female and male mice aged 6 to 12 weeks were used in the experiments due to animal welfare reasons. Mice of both sexes were randomly assigned to experimental groups. No sex-associated phenotypical differences were observed in the repeated experiments.

### Bacterial strains and culture condition

SL1344 as WT strain of *S*Tm [91] and derivatives of SL1344 were used in this study and are listed in S2 Table. SL1344 harboring chromosomal in-frame deletions were constructed by the lambda/red homologous recombination system [92] or P22 phage-mediated transduction, as described below. All strains were routinely grown overnight at 37˚C in Luria–Bertani (LB) broth with agitation or agar supplemented with 50 μg/ml streptomycin, or 10 μg/ml chloramphenicol, or 50 μg/ml kanamycin, or 100 μg/ml ampicillin, or 100 μg/ml spectinomycin, as required. As the indicated, bacteria were grown in M9 minimal media broth (47.7 mM Na$_2$HPO$_4$, 22 mM KH$_2$PO$_4$, 8.5 mM NaCl, 18.7 mM NH$_4$Cl, 1 mM MgSO$_4$, 0.1 mM CaCl$_2$, 0.1% glucose, and 0.01% histidine) supplemented with appropriated antibiotics.

### Mice

C57BL/6 mice were held under specific pathogen free (SPF) conditions at the institute of experiments of animals at the School of Pharmacy, Kitasato University. As necessary, SPF mice on the C57BL/6 background were purchased from Japan SLC. All mice were regularly maintained on the conventional rodent chow CE-2 (CLEA Japan).

### Construction of *S*Tm mutant strains

To create the isogenic deletion mutants, the target genes in the SL1344 genome were replaced with an antibiotic resistance cassette by lambda/red recombination system [92]. Primers with 40 flanking base pairs of the gene of interest and 20 base pairs of the chloramphenicol or kanamycin resistance cassette from pKD3 or pKD4 were designed, and then PCR was conducted to amplify the DNA fragment including the antibiotic resistance cassette flanked by the flanking regions of the gene of interest using pKD3 or pKD4 as a DNA template. The resulting PCR products were purified using the Monarch PCR & DNA cleanup kit (New England Biolabs) or FastGene gel/PCR extraction kit (NIPPON Genetics). *S*Tm harboring pKD46 were grown at 30˚C for 3 h in 50 ml of LB supplemented with ampicillin and 10 mM L-arabinose, and then bacterial cells were washed with ice-cold dH$_2$O and concentrated by resuspending in 10% glycerol. The resulting cells were transformed with 5 μl of the purified PCR products via electroporation at 1.8 kV for 5 ms, and then recovered in warm LB for 1.5 h at 37˚C along shaking and plated on LB plates containing 10 μg/ml chloramphenicol or 50 μg/ml kanamycin. The resulting colonies were purified on LB plate without antibiotics at 42˚C to cure cells of the pKD46. Desirable mutations were verified by PCR screening and then P22 phage-mediated transduction was conducted in SL1344. To generate double or multiple gene-disrupted mutants, P22 phage-mediated transduction was performed.

### Construction of complementary plasmid encoding *potAB* and *potFGHI*

*S*Tm *potAB* or *potFGHI* DNA region were amplified by PCR using primer set SL potA-BamHI-FW and SL potB-SphI-RV or SL potF-BamHI-FW and SL potI-SphI-RV (S3 Table). The

PCR products were digested with BamHI and SphI, and ligated into the same sites of pMW118, yielding pMW-*potAB* or pMW-*potFGHI*.

## Mouse infections

For *Salmonella*-induced colitis and systemic infection, the streptomycin mouse model was employed as described previously [32,93]. Briefly, 25 mg of streptomycin was administered by oral gavage to the mice 24 h prior to infection, and $5 \times 10^7$ CFU *S*Tm was infected by oral gavage or on coinfection experiments with 1:1 mixture of the bacterial culture (total $5 \times 10^7$ CFU). Alternatively, $1 \times 10^5$ CFU *S*Tm or 1:1 mixture of the bacterial culture (total $1 \times 10^5$ CFU) on coinfection was inoculated intraperitoneally to untreated mice. Mice were kept in cages with mesh floor to avoid transmission between mice. To determine *S*Tm colonization levels, fecal pellets, caecal contents, and spleen were freshly harvested and homogenized in sterile phosphate-buffered saline (PBS) (if necessary, with 0.5% Tergitol) using a Tissue Lyser devise (Qiagen) for 2 min at 25 Hz frequency. The homogenates were serially diluted in PBS and differentially plated on LB agar plates supplemented with appropriate antibiotic(s). After incubation at 37˚C overnight, colonies were counted and *S*Tm population sizes were calculated as CFU/g content or CFU/organ (spleen). The CI was calculated by the division of the population sizes of *S*Tm parent strains by those of their derivative mutants and normalized to the initial ratio in the inoculum. For survival assay, the infected mice were observed daily to evaluate survival.

## RNA isolation from bacteria and reverse transcription quantitative real-time PCR

Bacteria grown in LB or M9 media were isolated from the medium by centrifugation, and RNA was isolated using a Direct-zol RNA MiniPrep kit (Zymo Research) following the manufacturer's protocol. RNA was stored at –80˚C, and concentration and quality were determined spectrophotometrically using a NanoDrop 1000 Spectrophotometer (Thermo Fisher Scientific). Reverse transcription was performed using TaqMan Reverse Transcription reagents (Invitrogen). Quantitative real-time PCR (qPCR) was performed using SYBR Fast qPCR master mix (Kapa Biosystems) on CFX96 real-time PCR detection system (Bio-Rad) to amplify the target genes with specific primer pairs listed in S3 Table. Relative transcript levels were normalized to the *rpoD* gene and calculated by using the $2^{-\Delta CT}$ method [94].

## Measurement of polyamine content in culture supernatant

Bacteria were grown up to the middle-logarithmic growth phase in M9 supplemented with 3 mM putrescine or 3 mM spermidine. The bacterial culture was centrifuge at $20,000 \times g$ for 2 min, and 500 μl of the culture supernatant was isolated carefully, mixed with 50 μl of 100% trichloroacetic acid (FUJIFILM Wako Pure Chemical Corporation). The precipitates were removed, and the supernatant was filtered through a Cosmonice Filter W (Merck), and the polyamine concentration was analyzed using an HPLC system (Chromaster, Hitachi) equipped with a cation exchange column (#2619PH, $4.6 \times 50$ mm, Hitachi) as described previously [95].

## Lipocalin-2 ELISA

Fecal pellets were collected from mice and homogenized by bead beating and diluted in PBS. The resulting dilutions were assessed using the mouse lipocalin-2/NGAL ELISA DuoSet (R&D Systems) according to manufacturer's instructions.

## β-Galactosidase assay

β-Galactosidase assays were performed as previously described [96]. Shortly, bacteria were grown overnight in LB medium, subcultured 1/100 in 5 ml of LB medium, and then grown for 3 h at 37˚C in the same medium. One hundred microliters of the bacterial culture were mixed to 900 μl of Z buffer (60 mM $Na_2HPO_4·7H_2O$, 40 mM $NaH_2PO_4·H_2O$, 10 mM KCl, 1 mM $MgSO_4·7H_2O$, 50 mM β-mercaptoethanol), and then 20 μl of 0.1% (wt/vol) SDS and 40 μl of chloroform were added and mixed well. After preincubation for 5 min at 28˚C, 200 μl of *o*-nitrophenyl-β-D-galactopyranoside (ONPG) in 0.1 M potassium phosphate buffer (pH 7) (4 mg/ml) were added to start the reaction and developed at 28˚C. To stop the reactions, 500 μl of 1 M $Na_2CO_3$ were added, and the optical densities at 420 nm ($OD_{420}$) were measured. β-Galactosidase activity units (Miller units) were calculated as previously described [96].

## Preparation of secreted proteins

For preparation of secreted proteins through T3SS-1, *S*Tm were grown overnight in LB containing 0.3 M NaCl at 37˚C without aeration. In contrast, for preparation of secreted proteins that are dependent on T3SS-2, *S*Tm were grown in minimal medium LPM (pH 5.8) at 37˚C with aeration. For isolation of proteins secreted into culture supernatant, 1.5 ml of the supernatant from the respective bacteria culture was filtered, and trichloroacetic acid was added to the samples at a final concentration of 10%. After incubation on ice for 30 min, the samples were centrifuged at $16,000 \times g$ for 30 min, and the pellet was washed in ice-cold acetone. The resulting precipitated proteins were dissolved in SDS-PAGE sample buffer and subjected to SDS-PAGE and western blot analysis using specific antibodies.

## Assay of invasion of HeLa cells

*S*Tm grown in LB to the late logarithmic growth phase were infected into HeLa cells in Dulbecco's modified Eagle medium (DMEM) at a multiplicity of infection (MOI) of 10 bacteria per cell for 1 h at 37˚C with 5% $CO_2$. The cells were washed with PBS 3 times and added with DMEM containing 100 μg/ml gentamicin to kill the extracellular bacteria. After further incubation of 1 h, the cells were washed 3 times in PBS and then lysed with 1% Triton X-100. The samples that include intracellular bacteria were diluted and plated on LB agar plates to determine the numbers of cell-invaded bacteria.

## Contact hemolysis

Contact hemolysis by *S*Tm were done according to established protocol as previously described [33]. Shortly, *S*Tm were grown overnight in Brain Heart Infusion (BHI, BD Difco) medium at 37˚C along standing, and next day diluted at 1:100 in BHI medium. After further incubation of 3 h under the same condition, bacteria were harvested and resuspended in BHI medium ($1 \times 10^9$ CFU/ml). Sheep red blood cells (SRBCs) were washed 3 times in PBS and resuspended in BHI medium at 50% concentrations. Fifty microliter of bacterial suspension and 50 μl of 50% SRBC in BHI solution were added to wells of 96-well microtiter plates, and the plates were centrifuged at $2,000 \times g$ for 10 min, in which *S*Tm cells contact to SRBCs. After incubation at 37˚C for 2 h, samples were resuspended by adding 150 μl of PBS, and the plates were centrifuged at $2,000 \times g$ for 10 min. The supernatant (150 μl) was transferred into fresh 96-well microtiter plates, monitored for the presence of released hemoglobin at $OD_{542}$. Hemolytic activities were calculated as $100 \times (A_{542}$ experimental release$-A_{542}$ spontaneous release) / $(A_{542}$ experimental release$-A_{542}$ spontaneous release), where "spontaneous release" is the

amount of hemoglobin release from uninfected cells and "maximum release" is the amount of hemoglobin released uninfected but 1% Triton X-100 added cells.

## Assay of survival and replication within RAW264.7 cells

In case of T3SS-1-dependent survival, *S*Tm grown in LB to the late logarithmic growth phase was infected to RAW264.7 cell. In contrast, RAW264.7 cells were infected with *S*Tm grown overnight in LB for T3SS-2-dependent survival. Thus, *S*Tm infected RAW264.7 cells at an MOI of 10 and centrifuged at $1,000 \times g$ for 5 min to contact the bacteria and cells. *S*Tm-attached cells were incubated at 37˚C with 5% $CO_2$ for 25 min, washed twice in PBS, and incubated with 20 μg/ml gentamicin for 1.5 h under the same condition to kill the extracellular bacteria. After the incubation, the cell culture medium was removed carefully, and the cells were lysed with 1% Triton X-100. The samples were diluted and plated on LB agar plates to determine the numbers of engulfed bacteria. To determine the numbers of intracellular survived and replicated bacteria, gentamicin as described above was added for a period of 15 h 30 min (T3SS-1-dependent survival) or 19 h 30 min (T3SS-2-dependent survival). Fold increase as bacterial replication was presented as the mean ratio of bacteria recovered at 16 h or 20 h postinfection relative to the number of bacteria recovered after 2 h of infection containing 1.5 h of gentamicin treatment, defined as 1.

## LDH release assay

Cytotoxicity was evaluated by measuring the amount of released LDH. LDH release was determined using CytoTox 96 cytotoxicity assay kit (Promega) following manufacturer's protocol. Briefly, supernatant from RAW264.7 cells was harvested and evaluated for the presence of cytoplasmic enzyme LDH. The relative LDH release was calculated as $100 \times (A_{490}$ experimental release$-A_{490}$ spontaneous release$) / (A_{490}$ experimental release$-A_{490}$ spontaneous release$)$, where "spontaneous release" is the amount of LDH release from uninfected cells and "maximum release" is the amount of LDH released uninfected cells by addition of 1% Triton X-100.

## Effector translocation assay

Translocation of an effector-CyaA fusion protein via T3SS into host cells was evaluated by measuring the intracellular levels of cAMP using Direct cyclic AMP enzyme immunoassay kit (Arbor Assays). *S*Tm strains harboring the plasmids expressing effector-CyaA fusion proteins were infected to HeLa cells for 2 h or RAW264.7 cells for 16 h at an MOI of 10. Infected cells were lysed, and intracellular cAMP levels were determined as instructed by the manufacturer.

## Purification of the needle complex of T3SS-1 and electron microscopy

The *S*Tm Δ*fliGHI* and Δ*fliGHI* Δ*speABCEDF* Δ*potAB* Δ*potFGHI* cells were grown overnight in 30 ml of L-Na300 broth [1.0% (w/v) Bacto tryptone, 0.5% (w/v) Yeast extract, 300 mM NaCl] in a shaker at 37˚C. A 13 ml volume of overnight culture was added a 1.3 l volume of L-Na300 broth, and the *S*Tm cells were grown at 37˚C with shaking until the cell density had reached an $OD_{600}$ of ca. 1.0. After centrifugation ($10,000\ g$, 10 min, 4˚C), harvested cells were suspended in 60 ml of ice-cold sucrose solution (0.1 M Tris-HCl (pH 8.0), 0.5 M sucrose) in an ice bath. A 3.0 ml volume of lysozyme solution (10 mg/ml lysozyme in the sucrose solution) and a 0.6 ml volume of 0.1 M EDTA-NaOH (pH 8.0) were added to the cell suspension and stirred for 1 h at 4˚C, followed by adding n-dodecyl β-D-maltoside (DDM) and $MgSO_4$ at a final concentration of 1% (w/v) and 2 mM, respectively. To completely digest the chromosomal DNA, 5U of DNase (Roche) was added. After stirring overnight at 4˚C, the cell lysate was centrifuged ($10,000\ g$, 20 min, 4˚C) to remove cell debris. The supernatant was adjusted to pH 10.5 with 5

M NaOH, stirred at 4°C for 2 h, and centrifuged (10,000 *g*, 20 min, 4°C). After ultracentrifugation (94,000 *g*, 2 h, 4°C), pellets were resuspended in 1.5 ml of 10 mM Tris-HCl (pH 8.0), 5 mM EDTA, 1% (w/v) DDM, and this solution was loaded a 20% to 50% (w/w) sucrose density gradient in 10 mM Tris-HCl (pH 8.0), 5 mM EDTA, 1% (w/v) DDM. After ultracentrifugation (49,100 *g*, 16 h, 4°C), fractions containing T3SS proteins were collected and ultracentrifuged (90,000 *g*, 1 h, 4°C). Pellets were suspended in 20 μl of 20 mM Tris-HCl (pH 8.0), 1 mM EDTA, 50 mM NaCl, and 0.05% (w/v) DDM. Samples were applied to carbon-coated copper grids and negatively stained with 2% (w/v) uranyl acetate. Electron micrographs were taken using JEM-1400Flash (JEOL, Tokyo, Japan) operated at 100 kV.

### Bacterial clumping assay

To investigate whether PrgI is secreted and polymerized, the clumping assay was carried out by using Δ*invJ* genetic background strains [97]. Bacteria were grown in LB broth by incubation at 37°C along with standing. Optical densities of 660 nm of the culture were measured at both pre- and post-vortexing. For enhancement of clumping, nonmotile by deletion of *fliGHI* genes (Δ*fliGHI*) and T3SS-1-induction by plasmid-based inducible expression of *hilD* (pBAD-*hilD*) were employed.

### Construction of chromosomal transcriptional *lacZ* reporter strains

DNA fragment including the *prgH* promoter region were amplified by PCR using primer set pro-prgH-SalI and prgH-Rev-BamHI (S3 Table), and ligated into the same sites of pLD-*lacZ*Ω containing promoter less *lacZ* [44], yielding pLD-*prgHZ*. The *prgH::lacZ* allele was transduced into SL1344 via the P22 phage.

### Construction of plasmid expressing His- and HA-tagged T3SS protein

*S*Tm *hilD* DNA region were amplified by PCR using primer set SL hilD-SacI-FW and SL hilD-EcoRI-RV (S3 Table). The PCR products were digested with SacI and EcoRI, and ligated into the same sites of pBAD/His A (Invitrogen), yielding pBAD-*hilD*. *S*Tm *invJ* DNA region were amplified by PCR using primer set Pro-invJ-XhoI-F and invJ-BamHI-R (S3 Table). The PCR products were digested with XhoI and BamHI, and ligated into the same sites of pACHS-2HA [98], generating pACHS-InvJ-2HA, expression InvJ-2HA. *S*Tm *prgH* or *prgHI* DNA region were amplified by PCR using primer set Pro-prgH-SalI and prgH-BamHI-R or prgI--BamHI-R (S3 Table). The PCR products were digested with SalI and BamHI, and ligated into the XhoI and BamHI sites of pACHS-2HA [98], generating pACHS-PrgH-2HA and pACHS-PrgHI-2HA, expressing PrgH-2HA and PrgI-2HA, respectively. *S*Tm *ssaG* DNA region were amplified by PCR using primer set Pro-ssaG-XhoI and Rev-ssaG-BamHI (S3 Table). The PCR products were digested with XhoI and BamHI, and ligated into the same sites of pACHS-2HA [98], generating pACHS-SsaG-2HA, expressing SsaG-2HA.

### Construction of complementary plasmid expressing MBP-PrgH fusion protein

*S*Tm *prgH* DNA region were amplified by PCR using primer set SL prgH-EcoRI and SL prgH-HindIII-RV (S3 Table). The PCR products were digested with EcoRI and HindIII, and ligated into the same sites of pMAL-c2x, yielding pMAL-*prgH*.

### Measurement of polyamine content in murine feces and spleen

Murine feces and spleen were harvested in PBS and homogenized by bead beating. Polyamines were extracted by incubating in 5% trichloroacetic acid at 90°C for 30 min. Supernatants were

isolated by centrifugation and analyzed using a Jasco HPLC system with an InertSustain C18 column (particle size, 3 μm; internal diameter, 4.6 mm; length, 150 mm; GL Science) to determine polyamine levels according to the method as previously described [99]. Polyamines were separated using Buffer A (0.1 M sodium acetate (Wako Pure Chemical Corporation), pH 5.3 and 10 mM sodium octanesulfonate (Wako Pure Chemical Corporation) and Buffer B (0.1 M sodium acetate), pH 5.3 and 10 mM sodium octanesulfonate, 50% acetonitrile (Wako Pure Chemical Corporation) with the flow rate of 1.0 ml/min. The gradient program was set as follows: 0 to 10 min, A: 80% to 50%, B: 20% to 50%; 10 to 17 min, A: 50% to 40%, B: 50% to 60%; 17 to 18 min, A: 40% to 20%, B: 60% to 80%; 18 to 20 min, A: 20% to 80%, B: 80% to 20%; 20 to 25 min, A: 80%, B: 20%. Polyamines were post-column derivatized by mixing with OPA buffer consisting of 0.4 M boric buffer (pH 10.4, Wako Pure Chemical Corporation), 0.1% Brij-35 (Thermo Scientific), 2.0 ml/L 2-mercaptoethanol (Wako Pure Chemical Corporation), and 0.06% *o*-phthalaldehyde (Wako Pure Chemical Corporation) at 50˚C. The flow rate of derivatizing buffer was 0.4 ml/min and fluorescence was measured at an excitation wavelength of 340 nm and an emission wavelength of 455 nm.

### RNA isolation from murine spleen and reverse transcription quantitative real-time PCR

Murine spleen was harvested and stored in Buffer RLT (Qiagen) at −80˚C and then homogenized by bead beating. RNA from murine spleen was extracted using a RNeasy mini kit (Qiagen) following the manufacturer's protocol. RNA was stored at −80˚C, and concentration and quality were determined spectrophotometrically using a NanoDrop 1000 Spectrophotometer (Thermo Fisher Scientific). Reverse transcription was performed using TaqMan Reverse Transcription reagents (Invitrogen). Quantitative real-time PCR (qPCR) was performed using SYBR Fast qPCR master mix (Kapa Biosystems) on CFX96 real-time PCR detection system (Bio-Rad) to amplify the target genes with specific primer pairs listed in S3 Table. Relative transcript levels were normalized to the *ACTB* gene and calculated by using the $2^{-\Delta CT}$ method [94].

### Assay of arginase activities

Arginase activities were evaluated using Arginase activity assay kit (Cosmo Bio) according to manufacturer's instructions. Briefly, murine spleen was harvested in 0.1% Triton X-100 containing protease inhibitor cocktail on day 4 after infection with the indicated *S*Tm strains and homogenized by bead beating. Urea concentrations in the homogenates were determined and arginase activities were calculated based on following formula: arginase activities (unit/ml) = urea concentration (μg/ml) $\times 10^6$ / (60.06 (urea molecular weight) $\times$ 60 (min) $\times$ 20 (μl)).

### Amino acid content measurement

Sample preparation performed by extraction with 1% picric acid from murine spleen [100], followed by using EZ:faast (Phenomenex) according to the manufacturer's instruction with slight modifications. Amino acid contents in the sample were determined by LC-MS/MS (HPLC: Prominence system, Shimadzu, Japan; MS/MS: LCMS-8030, Shimadzu).

### Statistical analysis

Statistical tests were performed using GraphPad Prism 5 or 10 for MacOS (GraphPad Software). Statistical significance was determined by one-way ANOVA followed by Dunnett's multiple comparisons test, Mann–Whitney U test, a one-side Wilcoxon matched-pairs signed

rank test, Student *t* test, or Log-rank (Mantel–Cox) test. *P* values of less than 0.05 were considered statistically significant (*, $P < 0.05$; **, $P < 0.01$; ***, $P < 0.001$).

## Supporting information

**S1 Fig. The potAB mutation has no polar effect.** (A) Schematic information of loci of *spe* (yellow) and *pot* (orange). (B) Transcripts of *sifA* gene in the indicated *S*Tm strains grown in LPM media are presented as relative amounts to WT. Data are shown as the means ± standard deviations of the results from three independent experiments. ns, not significant; ***$P < 0.001$; Student *t* test. (C) In vitro growth in LB broth of *S*Tm WT (blue), Δ*speABCEDF* (red), Δ*potAB* Δ*potFGHI* (sky blue), Δ*speABCDEF* Δ*potAB* (magenta), Δ*speABCDEF* Δ*potFGHI* (light green), and Δ*speABCDEF* Δ*potAB* Δ*potFGHI* (orange) in LB. *n* = 4, respectively. The data underlying this figure can be found in S1 Data.
(EPS)

**S2 Fig. Introduction of plasmid encoding potAB restores the competitive colonization of the ΔspeABCEDF ΔpotAB ΔpotFGHI in the mouse model of colitis and systemic infection.** (A, B) Streptomycin-pretreated C57BL/6 mice were infected by oral gavage with the indicated 1:1 mixtures of *S*Tm strains. Feces, caecal content, and spleen were collected 4 days after infection to determine the competitive index (CI). CI values were determined by calculating bacterial number in feces, caecal content, and spleen. Bars represent median values, and *n* is indicated by the number of dots. ns, not significant; *$P < 0.05$; ***$P < 0.001$; one-way ANOVA followed by Dunnett's multiple comparisons test. The data underlying this figure can be found in S1 Data.
(EPS)

**S3 Fig. Polyamine uptake is required for functional expression of T3SS-1.** (A) Schematic of *sicA-sipBCDA* operon. HilA (gray), a master regulator of T3SS-1, activates transcription of *sicA* (green)-*sipBCDA* (yellow) operon. (B) Transcription levels of *sicA* gene are presented as units of β-galactosidase activities. Bars represent mean ± SD of the results from 5 independent experiments. ***$P < 0.001$; Student *t* test. (C) Secretion of T3SS-1 substrates, SipB and SipC, by the WT, Δ*invG* and Δ*speABCEDF* Δ*potAB* Δ*potFGHI* was assessed by western blot analysis with anti-SipB and anti-SipC antiserum. (D, E) Invasion of *S*Tm strains into HeLa cells. Invasive capacity is presented as relative invasion to WT. Bars represent means ± SD of the results from at least 6 independent experiments. ns, not significant; ***$P < 0.001$; one-way ANOVA followed by Dunnett's multiple comparisons test. (F) Representative images of released hemoglobin by contact hemolysis. (G) T3SS-1-dependent contact hemolytic activity of *S*Tm strains. Hemolytic activity is presented as relative hemolysis to WT. Bars represent mean ± SD of the results from at least 4 independent experiments. ns, not significant; ***$P < 0.001$; one-way ANOVA followed by Dunnett's multiple comparisons test. (H) T3SS-1-dependent cytotoxic activity of *S*Tm strains was determined by measuring the amounts of released LDH into the supernatant. Bars represent means ± SD of the results from 4 independent experiments. *$P < 0.05$; ***$P < 0.001$; one-way ANOVA followed by Dunnett's multiple comparisons test. (I) T3SS-1-dependent survival and replication of *S*Tm strains in RAW264.7 cells. RAW264.7 cells were infected with the indicated *S*Tm strains for 16 h, and the ability to survival and replicated within RAW264.7 cells were determined as fold increase. Bars represent means ± SD of the results from at least 6 independent experiments. ***$P < 0.001$; one-way ANOVA followed by Dunnett's multiple comparisons test. (J) Translocation of SopD-CyaA fusion protein into HeLa cells in the T3SS-1-dependent fashion. Monolayers of HeLa cells were infected with the indicated *S*Tm strains expressing SopD-CyaA fusion protein for 2 h. Bars represent

means ± SD from 3 independent experiments. ns, not significant; ***$P < 0.001$; one-way ANOVA followed by Dunnett's multiple comparisons test. The data underlying this figure can be found in S1 Data.
(EPS)

**S4 Fig. Effect of potAB overexpression in bacterial growth.** In vitro growth of *S*Tm WT, Δ*speABCEDF* Δ*potAB* Δ*potFGHI* harboring pMW118, and Δ*speABCEDF* Δ*potAB* Δ*potFGHI* harboring pMW-*potAB* in M9 media-supplemented spermidine or without as indicated. Growth was determined by measuring $OD_{660}$. $n = 3$, respectively. The data underlying this figure can be found in S1 Data.
(EPS)

**S5 Fig. Polyamine homeostasis is required for the translocation of SopD-CyaA fusion protein.** (A, B) Translocation of T3SS-1 effector. Monolayers of HeLa cells were infected with the indicated *S*Tm strains expressing SopD-CyaA fusion protein for 30 min. Cell associated bacteria (A) and intracellular cAMP levels (B) were determined. Bars represent means ± SD of the results from 3 independent experiments. ns, not significant; *$P < 0.05$; **$P < 0.01$; ***$P < 0.001$; one-way ANOVA followed by Dunnett's multiple comparisons test. The data underlying this figure can be found in S1 Data. (C) Intracellular expression of SopD-CyaA and DnaK in the indicated *S*Tm strains by western blot analysis using anti-CyaA and anti-DnaK antibodies.
(EPS)

**S6 Fig. Spermidine uptake is involved in functional expression of T3SS-2.** (A) Transcripts of T3SS-2 genes (*sseB*, *ssaG*, and *sseJ*) from *S*Tm strains grown in LPM are presented as relative expression to WT. Bars represent means ± SD of the results from 3 independent experiments. ns, not significant; *$P < 0.05$; **$P < 0.01$; ***$P < 0.001$; one-way ANOVA followed by Dunnett's multiple comparisons test. (B) Secretion of a T3SS-2 substrate, SseB. Secreted and intracellular SseB proteins were analyzed by western blot analysis with anti-SseB antiserum. (C) Relative engulfment. $n = 3$, respectively. ns, not significant; ***$P < 0.001$; one-way ANOVA followed by Dunnett's multiple comparisons test. (D, H) T3SS-2-dependent survival and replication of *S*Tm strains within RAW264.7 cells. RAW264.7 cells were infected with the indicated *S*Tm strains for 20 h, and the ability to survival and replicated within RAW264.7 cells were determined as fold increase. As indicated, 4MCHA, an inhibitor of spermidine synthase, was added into cell culture media at a concentration of 1.2 mM. Bars represent means ± SD of the results from at least 3 independent experiments. ns, not significant; *$P < 0.05$; **$P < 0.01$; ***$P < 0.001$; one-way ANOVA followed by Dunnett's multiple comparisons test. (E–G) T3SS-2-dependent cytotoxic activity of *S*Tm strains was determined by measuring the amounts of released LDH into the supernatant. Data also are presented as relative cytotoxicity to WT (G) and are shown as the means ± SD of the results from at least 4 independent experiments. ns, not significant; *$P < 0.05$; **$P < 0.01$; ***$P < 0.001$; one-way ANOVA followed by Dunnett's multiple comparisons test or Student $t$ test. The data underlying this figure can be found in S1 Data. (I) Intracellular expression of SseJ-CyaA and DnaK in the indicated *S*Tm strains by western blot analysis using anti-CyaA and anti-DnaK antibodies.
(EPS)

**S7 Fig. PrgI is the needle subunit of T3SS-1 needle complex.** (A) Proportion of needle complexes harboring the needle filament from the WT or Δ*prgI* or Δ*speABCEDF* Δ*potAB* Δ*potFGHI* grown in LB supplemented with spermidine or without supplementation. Analyzed particles: WT = 390; WT + spermidine = 390; Δ*prgI* = 286; Δ*prgI* + spermidine = 287; Δ*speABCEDF* Δ*potAB* Δ*potFGHI* = 370; Δ*speABCEDF* Δ*potAB* Δ*potFGHI* + spermidine = 358 from 2

independent experiments. ND, not detected. The data underlying this figure can be found in S1 Data. (B–G) Representative electron micrographs of negatively stained needle complexes isolated from the WT or Δ*prgI* or Δ*speABCEDF* Δ*potAB* Δ*potFGHI* grown in LB supplemented with spermidine or without supplementation. Scale bar = 100 nm.
(EPS)

**S8 Fig. Expression of the T3SS needle subunit.** (A) Schematic of *prgHIJK-orgABC* operon. The master regulator HilA (gray) activates the transcription of *prgHIJK-orgABC* operon, containing the *prgI* (green) encoding the T3SS-1 needle subunit. (B) Transcription levels of *prgH* gene are presented as units of β-galactosidase activities. Bars represent means ± SD of the results from 8 independent experiments. ns, not significant; ***$P < 0.001$; one-way ANOVA followed by Dunnett's multiple comparisons test. The data underlying this figure can be found in S1 Data. (C–E) Expression of the HA-tagged T3SS proteins, the inner ring protein (PrgH) of T3SS-1, the needle subunit (PrgI) of T3SS-1, and the needle subunit (SsaG) of T3SS-2. Western blot analysis was done by using anti-HA and anti-DnaK antibody.
(EPS)

**S9 Fig. Assembly of the PrgI needle subunits is required for secretion and translocation of T3SS substrates.** (A, B) Secretion of the middle substrate (SipB) and the late substrates (SopD) from the indicated *S*Tm strains was assessed by western blot analysis with anti-SipB antiserum and anti-HA antibody. (C) Translocation of SopD-CyaA fusion protein into HeLa cells in the PrgI-dependent fashion. Monolayers of HeLa cells were infected with the indicated *S*Tm strains expressing SopD-CyaA fusion protein for 2 h. (D) Invasion of *S*Tm strains into HeLa cells. Invasive capacity is presented as relative invasion to WT. Bars represent means ± SD from at least 3 independent experiments. ***$P < 0.001$; Mann–Whitney U test. The data underlying this figure can be found in S1 Data.
(EPS)

**S10 Fig. Polyamine contents in RAW264.7 cells treated with DFMO.** Polyamine levels of RAW264.7 cells treated with DFMO. Bars represent median values, and *n* is indicated by the number of dots. ns, not significant; **$P < 0.01$; Mann–Whitney U test. The data underlying this figure can be found in S1 Data.
(EPS)

**S11 Fig. Link between gut inflammation and polyamine levels in DSS model mouse.** (A, B) Polyamine levels in feces (A) and spleen (B) of DSS model mice of colitis and untreated control mice. Bars represent median values, and *n* is indicated by the number of dots. *$P < 0.05$; **$P < 0.01$; Mann–Whitney U test. The data underlying this figure can be found in S1 Data.
(EPS)

**S12 Fig. Arginase activity and amino acid contents in mice infected with *S*Tm.** (A) Arginase activity of the spleen homogenate in mice infected with the *S*Tm WT or Δ*invG* Δ*ssaV* and uninfected mice ($n = 3$, respectively). Data are shown as the means ± standard deviations of the results from 3 independent experiments. ns, not significant; *$P < 0.05$; **$P < 0.01$; one-way ANOVA followed by Dunnett's multiple comparisons test. (B) Amino acids content in the spleen homogenate in the *S*Tm WT-infected and uninfected ($n = 3$, respectively). Data are shown as the means ± standard deviations. ns, not significant; *$P < 0.05$; Student *t* test. The data underlying this figure can be found in S1 Data.
(EPS)

**S13 Fig. Polyamine contents in mice with spermidine drinking.** (A, B) Polyamine levels in feces (A) and spleen (B) of mice supplemented with spermidine in the drinking water or tap

water, a control. Bars represent median values, and *n* is indicated by the number of dots. ns, not significant; **$P < 0.01$; Mann–Whitney U test. The data underlying this figure can be found in S1 Data.
(EPS)

**S14 Fig. Model for the exploitation of host polyamines in STm infection.** In the gut, *S*Tm imports exogenous polyamines, which required for the functional expression of T3SS-1 and thereby invades intestinal epithelial cells (IECs) and elicits the gut inflammation, leading to luminal expansion of *S*Tm. Thereafter, in *Salmonella*-containing vacuole, polyamine uptake by *S*Tm is also needed to express the functional T3SS-2. Polyamine contributes to the assembly of needle in both T3SS-1 and -2 machineries, allowing to inject effectors into host cytoplasm. On the other hand, *S*Tm infection increases the expression of host arginase, resulting in elevation of polyamine contents. This also shifts M0 macrophage to M2 macrophage, which creates a favorable environment for *S*Tm survival. In summary, the arginase–polyamine immunometabolism signalling axis is a critical driver for *S*Tm infection. Created with BioRender.com (Agreement number: FX26WV16DO).
(EPS)

**S1 Data. Excel spreadsheet containing, in separate sheets for each figure, the underlying and individual numerical data used for Figs** 1B–1G, 2A–2F, 3A–3H, 4A–4C, 5A–5C, 6C, 7A–7D, 8B–8G, S1B, S1C, S2A, S2B, S3B, S3D, S3E, S3G–S3J, S4, S5A, S5B, S6A, S6C–S6H, S7A, S8B, S9C, S9D, S10, S11A, S11B, S12A, S12B, S13A, **and** S13B.
(XLSX)

**S1 Table. Nomenclature of type 3 secretion components.**
(PDF)

**S2 Table. Bacterial strains used in this study.**
(PDF)

**S3 Table. Oligonucleotide primers used in this study.**
(PDF)

**S1 Raw Images. Uncropped pictures used in Figs** 6D–6G, S3C, S5C, S6B, S6I, S8C–S8E, S9A and S9B.
(PDF)

## Acknowledgments

We thank Yuya Kishino, Mai Horino, Miki Kobayashi, Yukina Nakamura, Moka Ebinuma, Nanako Kimura, Shu Komura, Momoko Kobayashi, and Natsuki Watanabe for technical assistance.

## Author Contributions

**Conceptualization:** Tsuyoshi Miki.

**Funding acquisition:** Tsuyoshi Miki, Miki Kinoshita, Takeshi Haneda, Tohru Minamino, Yun-Gi Kim.

**Investigation:** Tsuyoshi Miki, Takeshi Uemura, Miki Kinoshita, Yuta Ami, Shin Kurihara, Takeshi Haneda, Tohru Minamino.

**Resources:** Masahiro Ito, Nobuhiko Okada, Takemitsu Furuchi, Takeshi Haneda.

**Validation:** Tsuyoshi Miki, Takeshi Uemura, Miki Kinoshita, Shin Kurihara, Tohru Minamino.

**Writing – original draft:** Tsuyoshi Miki, Takeshi Uemura, Tohru Minamino.

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
