## [Editor Report · Decision Letter 0]

17 Jan 2024

Dear Dr Miki, 

Thank you for submitting your manuscript entitled "Salmonella exploits host polyamines for assembly of the type 3 secretion machinery" for consideration as a Research Article by PLOS Biology.

Your manuscript has now been evaluated by the PLOS Biology editorial staff, as well as by an academic editor with relevant expertise, and I am writing to let you know that we would like to send your submission out for external peer review.

Once your full submission is complete, your paper will undergo a series of checks in preparation for peer review. After your manuscript has passed the checks it will be sent out for review. To provide the metadata for your submission, please Login to Editorial Manager (https://www.editorialmanager.com/pbiology) within two working days, i.e. by Jan 19 2024 11:59PM.

Kind regards,

Melissa

Melissa Vazquez Hernandez, Ph.D.

Associate Editor

PLOS Biology

---

## [Decision Letter · Decision Letter 1]

29 Feb 2024

Dear Dr Miki,

Thank you for your patience while your manuscript "Salmonella exploits host polyamines for assembly of the type 3 secretion machinery" was peer-reviewed at PLOS Biology. It has now been evaluated by the PLOS Biology editors, an Academic Editor with relevant expertise, and by several independent reviewers. I am deeply sorry about the lenght of the review process.

In light of the reviews, which you will find at the end of this email, we would like to invite you to revise the work to thoroughly address the reviewers' reports. As you will see below, the reviewers are broadly positive about your study; however they raise some concerns that should be addressed for further consideration at PLOS Biology. All reviewers have recommended conducting additional experiments to solidify the conclusions, including an infection kinetic in early stages, measuring expression levels of T3SS components and evaluating whether the uptake mutants can invade the host. Reviewer #2 and #3, in particular, have raised concerns about the specific effects of polyamines on the T3SS needle. Additionally, all reviewers wonder what the effects of polyamine excess in the host were. After discussion with the Academic Editor and the reviewers, we have determined that addressing the concerns raised by all reviewers is necessary for your manuscript to be considered for publication in PLOS Biology.

Given the extent of revision needed, we cannot make a decision about publication until we have seen the revised manuscript and your response to the reviewers' comments. Your revised manuscript is likely to be sent for further evaluation by all or a subset of the reviewers.

**IMPORTANT - SUBMITTING YOUR REVISION**

*Re-submission Checklist*

*Published Peer Review*

*PLOS Data Policy*

*Blot and Gel Data Policy*

Sincerely,

Melissa

Melissa Vazquez Hernandez, Ph.D.

Associate Editor

PLOS Biology

REVIEWER's COMMENTS

Reviewer #1: 

The manuscript "Salmonella exploits host polyamines for assembly of the type 3 secretion machinery" describes the effect of polyamines on Salmonella virulence and hints on how Salmonella induces host polyamine production to better survive at the infection sites. The manuscript follows a double path, one addressing the effect of polyamine on the assembly of the T3SS-1 (possibly translatable to T3SS-2) and the other one dealing with intracellular survival at later stages of infection, mediated by upregulation of polyamine production into the host and Salmonella SPD uptake.

Overall this is a promising study with interesting new results, however especially in the first part (Fig 1-4) the authors reasoning is not always obvious and the two paths described above often blend into each other. It would probably be helpful to rethink how to present the first part, also when it comes to the nomenclature of the different mutants and the weight the authors give to polyamine synthesis vs uptake contribution to Salmonella infection. 

Specific comments:

1. The hypothesis that Salmonella could uptake polyamines through other transporters beside potABCD and potFGHI stems from the observation that PUT/SPD supplementation restores the growth defect in the ΔspeABCEDF ΔpotAB ΔpotFGHI mutant (Fig1). However there is no formal proof that this mutant can import polyamines and at what extent. It would be relatively easy to measure the bacterial intracellular polyamine concentration in this mutant grown with and without polyamine supplementation. This is particularly important since this mutant is then used as a proxy for polyamine uptake during in vivo infections. While this mutant partially recovers its colonization capacity when SPD is supplemented (Fig 2C-D), it is unclear at what extent this is due to SPD uptake, or to other reasons (e.g. stabilization of bacterial membrane; SPD effect on the host side). 

2. All mouse infections were analysed at 4 days pi. In the Salmonella streptomycin mouse model this is a rather late stage, when multiple cycles of infection into the intestinal epithelium and reseeding into the intestinal lumen have happened. Moreover, it is known that bacterial mutants impaired in polyamine biosynthesis/transport are sensitive to multiple stressors (e.g. pH, oxidative stress). With this in mind, it becomes unclear at which stage of infection polyamines might play a role in Salmonella colonization. An infection kinetic experiment which looks at earlier stages of infection would be helpful to tease apart how polyamine uptake would affect the infection progression. For example, does the ΔspeABCEDF ΔpotAB ΔpotFGHI mutant colonize the lumen at similar levels as the wt at earlier time points (for example 12 hours p.i.)? If this is the case, would the mutant be impaired at infecting the cecum tissue (as suggested by the defective T3SS-1 assembly)? 

3. How would polyamines facilitate the assembly of the T3SS-1? It would be interesting to at least discuss a possible mechanism. Moreover, considering that in a normal infection setting Salmonella arrives at the infection site with its own set of intracellularly synthesized polyamines, does the Salmonella transporter mutant ΔpotAB ΔpotFGHI also show signs of impaired T3SS-1 assembly? This mutant seems to have no colonization defect in the cecal content, but rather a defect in spreading to systemic sites. Would this suggest that polyamine uptake might have a more prominent role in bacterial intracellular survival and/or spread?

4. The results with DFMO supplementation are very interesting and potentially impactful for therapeutic treatments, considering this is an already approved drug. This aspect could probably be stressed out more in the abstract as a point of novelty. Also, is the polyamine content reduced in macrophages in the presence of DFMO? How would DFMO affect the macrophage differentiation status? Please note that there is a recent pre-print showing similar/overlapping results (https://www.biorxiv.org/content/10.1101/2023.09.29.560257v2)

5. The authors discuss the possibility that Salmonella induces macrophage differentiation towards an M2 status through upregulation of polyamines, which would facilitate Salmonella intracellular replication. In principle the authors have all necessary tools available to complement their studies by also testing this hypothesis, for example following Salmonella infection in primary macrophages. 

Reviewer #2: 

Polyamines are ubiquitous molecules with diverse, but poorly understood functions. Polyamines have been shown to both influence and be a target of bacterial infections. In this study, Miki and colleagues describe the role of bacterial polyamine levels in infection by Salmonella enterica, an influence on the two Salmonella type III secretion systems (T3SS), and an effect of Salmonella infection on host polyamine biosynthesis. Together, the results indicate that sufficient polyamine levels are required for infection and that Salmonella actively promote increased levels.

This is a nice manuscript at the interface of bacterial virulence and host response. The results are presented in a clean and logical manner. The striking effect of polyamine levels on infection and the manipulation of host polyamine synthesis (Fig. 1, 2, 3, 5, 6 and corresponding Suppl. Figures) are worked out clearly; however, similar results were already shown in earlier publications, including for Salmonella. The completely novel observation - the specific effect on T3SS needle formation - is less obviously convincing, see comments below.

Major comments:

1. There are several publications that have described links between host polyamines and bacterial infections, including for Salmonella. This previous knowledge is partially discussed in the discussion part, but should already be presented more clearly in the introduction. This includes, but is not limited to:

 o Salmonella infection increases Arg-1 and Arg-2 expression (refs. 49, 50, mentioned in lines 575-577)

 o Spermidine increases T3SS expression in Pseudomonas aeruginosa (DOIs 10.1128/spectrum.00644-22, 10.1371/journal.pone.0001291)

 o Polyamine levels strongly influence protein expression in Salmonella, including the T3SS components (ref. 23, mentioned in line 532)

 o A very recent paper by Nair et al. (DOI 10.1016/j.micres.2024.127605), which shows multilayered effect of polyamines on expression of virulence factors, including the T3SS

2. The specific lack of needle formation at low polyamine levels is a central point of the study. In the light of the effect of polyamines on expression levels of many T3SS components, the expression levels of SPI-1 and SPI-2 components, especially the needle subunit, should be tested under the conditions used for the experiments. This would also help to determine the reason for the discrepancies with earlier studies (ref. 23 and Nair et al., see previous point).

 o Relatedly, can the authors discriminate between assembly of the needle and secretion of the needle subunit? If the effect is on the secretion, is it possible that an energy deficiency, caused by lack of polyamines and also influencing groth (see Fig. 1) is the cause?

3. It is rather surprising that the authors observe secretion of middle and late T3SS substrates in the absence of T3SS needles. It was shown that in the absence of the needle, the T3SS secretin adopts a closed state, most likely sealing the T3SS (Hu et al, DOI 10.1038/s41564-019-0545-z). Similarly, strains lacking the needle subunit do not export any T3SS substrate in several bacteria where this was tested. To better understand this unexpected phenotype, which is central to the manuscript, strains lacking the needle component should be used as a control in the respective experiments.

 o Relatedly, is it possible that both strains used form complete T3SS, but lose the needle in lower polyamine conditions, either over time or during the purification procedure? In either case, this central aspect (Fig. 4A) should be supported by bigger fields of view (perhaps as Suppl. Data) and quantification of purified needle complexes with and without needle in both strains.

4. Overexpression of potAB and addition of polyamines much more than complements the effect of the used deletions in many experiments (e.g, Fig. 3G, S3D, S3E, S3F, S3J, S4E). This implies that bacterial access to polyamines is not only required, but actually limiting for infection. It also raises the question why bacteria don't express higher levels of potAB in infection settings. It would be highly relevant and interesting to expand on this finding.

Alternatively, it might just mean that access to polyamines leads to increased bacterial growth, which then leads to stronger downstream effects. Growth of the motAB overexpression strain should therefore be tested (like in Fig. 1C-F).

5. Lines 493/494, "These results clearly indicate the causal link between increased SPD levels and enhanced Salmonella pathogenesis." - isn't it equally possible that increased levels of polyamines exert a stress on the host cells, which makes them more susceptible to infection? This would be an indirect, rather than a causal link.

6. In the results part, results on SPI-1 and SPI-2 need to be more clearly separated. They are two different systems used at very different circumstances. Currently, it is easy to overlook which system the authors discuss in various parts, e.g. in Fig. 3 and Fig. 4.

Minor points:

1. The common Sct T3SS nomenclature (DOI 10.1128/mmbr.62.2.379-433.1998 and 10.1007/82_2020_210) should be used at least in parallel for anyone not familiar with the SPI-1/2-specific names of T3SS components.

2. Can the authors speculate on the different (sometimes opposite) impact of the mutations on the colonization of feces/cecum vs. spleen?

3. Why was a HilD overepxression strain used in Fig. 4BC and how do the results look without HilD overexpression? 

4. The reduced colonizati

---

## [Decision Letter · Decision Letter 2]

3 Jun 2024

Dear Dr Miki,

Thank you for your patience while we considered your revised manuscript "Salmonella exploits host polyamines for assembly of the type 3 secretion machinery" for consideration as a Research Article at PLOS Biology. Your revised study has now been evaluated by the PLOS Biology editors, the Academic Editor and the original reviewers. 

As you will see, the reviewers are generally satisfied with the revision and the additional data included. However, Reviewer #1 would like you to discuss why spermidine does not restore T3SS needle assembly in the mutants, and Reviewer #2 requests that the secretion of middle and late T3SS substrates in the absence of the needle be evaluated experimentally. After discussion with the Academic Editor, we require both requests to be addressed as they are a central part of the manuscript.

In light of the reviews, which you will find at the end of this email, we are pleased to offer you the opportunity to address the remaining experimental requests from Reviewer #2 in a revision that we anticipate should not take you very long. We will then assess your revised manuscript and your response to the reviewers' comments with our Academic Editor aiming to avoid further rounds of peer-review, although might need to consult with the reviewers, depending on the nature of the revisions.

**IMPORTANT - SUBMITTING YOUR REVISION**

*Resubmission Checklist*

*Published Peer Review*

*PLOS Data Policy*

*Blot and Gel Data Policy*

Sincerely,

Melissa

Melissa Vazquez Hernandez, Ph.D.

Associate Editor

PLOS Biology

REVIEWERS' COMMENTS

Reviewer #1: 

I thank the authors for their extensive response and considerations related to the comments given by the reviewers. From my point of view the comments were intensively addressed, discussed, and where indicated translated into an improved outline and wording of the manuscript. 

I only have one further comment: Do the authors have any explanation on why spermidine supplementation doesn't restore T3SS-1 needle assembly in the ΔspeABCEDF ΔpotAB ΔpotFGHI mutant (Fig S7A-F)? This should be addressed and discussed, since polyamines effect on T3SS assembly is the central/novel result of the manuscript.

Reviewer #2: 

The authors have addressed most of the requests and clarified many points, which further improves the manuscript.

However, in response the request to test the secretion of middle and late T3SS substrates in absence of the T3SS needle subunit, the authors present data that absence of needles decreases invasiveness and effector translocation, but not on secretion of middle and late substrates.

These experiments should be included and interpreted, given that the surprising finding that the needle-less ∆speABCEDF ∆potAB ∆potFGHI strain secrete middle and late substrates is central to the manuscript.

Reviewer #3: 

The revised version of this manuscript has answered my concerns and I think it is now ready to be published.

---

## [Editor Report · Decision Letter 3]

27 Jun 2024

Dear Dr Miki,

Thank you for your patience while we considered your revised manuscript "Salmonella exploits host polyamines for assembly of the type 3 secretion machinery" for publication as a Research Article at PLOS Biology. This revised version of your manuscript has been evaluated by the PLOS Biology editors, the Academic Editor.

Based on our Academic Editor's assessment of your revision, we are likely to accept this manuscript for publication. Please make sure to address the following data and other policy-related requests.

a) We routinely suggest changes to titles to ensure maximum accessibility for a broad, non-specialist readership, and to ensure they reflect the contents of the paper. In this case, we would suggest a minor edit to the title, as follows. Please ensure you change both the manuscript file and the online submission system, as they need to match for final acceptance.

"Salmonella Typhimurium exploits host polyamines for assembly of the type 3 secretion machinery"

Please supply the numerical values either in the a supplementary file or as a permanent DOI’d deposition for the following figures:

Figure 1BCDEFG, 2ABCDEF, 3ABCDEFGH, 4ABC, 5ABC, 6C, 7ABCD, 8BCDEFG, S1BC, S2AB, S3BDEGHIJ, S4, S5AB, S6ACDEFGH, S7A, S8B, S9CD, S10, S11AB, S12AB, S13AB

c) Please cite the location of the data clearly in all relevant main and supplementary Figure legends, e.g. “The data underlying this Figure can be found in S1 Data” or “The data underlying this Figure can be found in https://doi.org/10.5281/zenodo.XXXXX”

d) We require the original, uncropped and minimally adjusted images supporting all blot and gel results reported in the following figures:

Figure 6DEFG, S3C, S5C, S6BI, S8CDE, S9AB

We will require these files before a manuscript can be accepted so please prepare and upload them now. Please carefully read our guidelines for how to prepare and upload this data: https://journals.plos.org/plosbiology/s/figures#loc-blot-and-gel-reporting-requirements

e) Please ensure that your Data Statement in the submission system accurately describes where your data can be found and is in final format, as it will be published as written there.

f) Per journal policy, if you have generated any custom code during the curse of this investigation, please make it available without restrictions upon publication. Please ensure that the code is sufficiently well documented and reusable, and that your Data Statement in the Editorial Manager submission system accurately describes where your code can be found.

We expect to receive your revised manuscript within two weeks. 

*Published Peer Review History*

*Press*

Sincerely,

Melissa

Melissa Vazquez Hernandez, Ph.D.

Associate Editor

PLOS Biology

---

## [Editor Report · Decision Letter 4]

2 Jul 2024

Dear Dr Miki,

Thank you for the submission of your revised Research Article "Salmonella Typhimurium exploits host polyamines for assembly of the type 3 secretion machinery" for publication in PLOS Biology. On behalf of my colleagues and the Academic Editor, [**AE Name**], I am pleased to say that we can in principle accept your manuscript for publication, provided you address any remaining formatting and reporting issues. These will be detailed in an email you should receive within 2-3 business days from our colleagues in the journal operations team; no action is required from you until then. Please note that we will not be able to formally accept your manuscript and schedule it for publication until you have completed any requested changes.

PRESS

Sincerely, 

Melissa

Melissa Vazquez Hernandez, Ph.D., Ph.D.

Associate Editor

PLOS Biology
